# Codifying Character Logic in Role-Playing

**Letian Peng, Jingbo Shang**[*]
Department of Computer Science
University of California, San Diego
{lepeng, jshang}@ucsd.edu

## Abstract

This paper introduces **Codified Profiles** for role-playing, a novel approach that represents character logic as structured, executable functions for behavioral decision-making. Converted by large language model (LLM) from textual profiles, each codified profile defines a set of functions `parse_by_scene(scene)` that output multiple logic-grounded assertions according to `scene`, using both explicit control structures (e.g., if-then-else) and flexible `check_condition(scene, question)` functions where each `question` is a semantically meaningful prompt about the `scene` (e.g., *"Is the character in danger?"*) discriminated by the role-playing LLM as *true*, *false*, or *unknown*. This explicit representation offers three key advantages over traditional prompt-based textual profiles, which append character descriptions directly into text prompts: (1) *Persistence*, by enforcing complete and consistent execution of character logic, rather than relying on the model's implicit reasoning; (2) *Updatability*, through systematic inspection and revision of behavioral logic, which is difficult to track or debug in prompt-only approaches; (3) *Controllable Randomness*, by supporting stochastic behavior directly within the logic, enabling fine-grained variability that prompting alone struggles to achieve. To validate these advantages, we introduce a new benchmark constructed from 83 characters and 5,141 scenes curated from Fandom, using natural language inference (NLI)-based scoring to compare character responses against ground-truths. Our experiments demonstrate the significant benefits of codified profiles in improving persistence, updatability, and behavioral diversity. Notably, by offloading a significant portion of reasoning to preprocessing, codified profiles enable even 1B-parameter models to perform high-quality role-playing, providing an efficient, lightweight foundation for local deployment of role-play agents. [2]

## 1 Introduction

Role-playing [Chen et al., 2024a,b] has emerged as an iconic capability of modern large language models (LLMs) [OpenAI, 2023, Touvron et al., 2023], enabling them to simulate characters with distinct personas. This ability has spurred a surge of interest in both research [Shao et al., 2023, Wang et al., 2023a,b, Sadeq et al., 2024] and applications[3], ranging from emotional companions and storytelling assistants [Brahman et al., 2021, Qin et al., 2024] to non-playable game characters [Yu et al., 2025] and metaverse agents [Zhou, 2023, Lifelo et al., 2024].

Despite this progress, current LLM-based role-playing systems interpret character logic by appending character descriptions directly into prompts. This approach *lacks persistence and controllability*: character behavior is governed by the LLM's implicit reasoning, which is often brittle and inconsistent.

---

[*] Corresponding author.
[2] Codes and datasets are available at https://github.com/KomeijiForce/Codified_Profile_Koishiday_2025
[3] https://character.ai/

39th Conference on Neural Information Processing Systems (NeurIPS 2025).

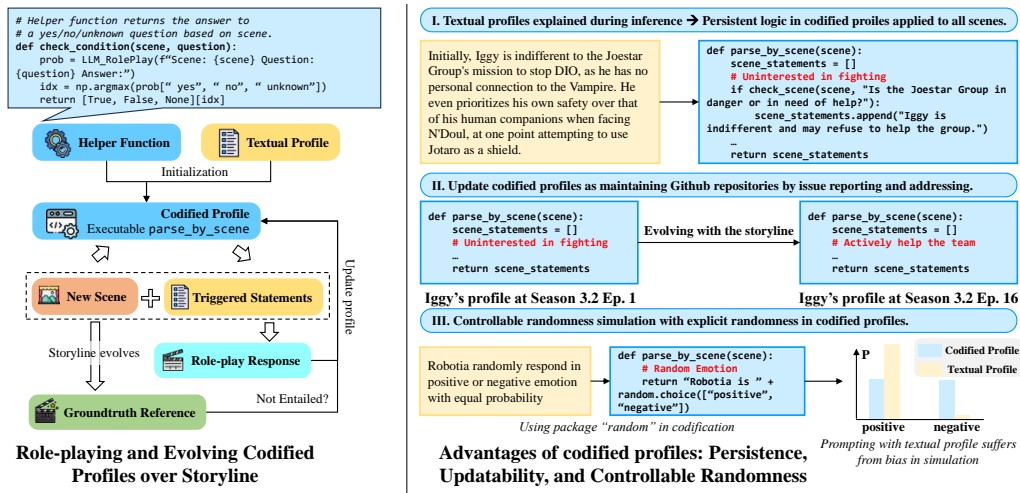

Figure 1: A presentation of the workflow and advantages of codified profiles.

Even well-crafted profiles frequently yield unpredictable results, particularly in complex or ambiguous scenes, and offer little transparency, making them *difficult to debug or update*. In addition, interpreting profile logic at runtime imposes significant computational overhead and introduces latency, limiting scalability and responsiveness.

To address these challenges, we propose **Codified Profiles**, which compile character logic from natural language descriptions into executable functions. Unlike prompt-based role-playing, which appends character descriptions to prompts and relies on runtime interpretation, codified profiles are constructed in advance. They define character behavior using explicit control logic, producing consistent and interpretable outputs while reducing reliance on the LLM's implicit reasoning. This design enhances both behavioral persistence and computational efficiency by eliminating the need for repeated logic interpretation during interaction.

As shown in Figure 1, we convert textual profiles into `parse_by_scene(scene)` functions via LLM prompting, which use control structures to generate logic-based `triggered_statements`. Each function includes multiple callable functions `check_condition(scene, question)` that evaluate scene-specific conditions using the role-playing LLM's logits over verbalizers (e.g., "yes", "no", "unknown"), enabling flexible semantic reasoning within a deterministic framework. At inference, the function outputs guide the LLM's response based on explicit character logic.

Codified profiles offer three key advantages over prompt-based methods. First, they ensure *persistence* by enforcing consistent execution of character logic, avoiding reliance on the LLM's implicit reasoning. Second, they enable *updatability*, as structured, executable logic can be systematically inspected, debugged, and revised. Third, they support *controllable randomness*, allowing explicit encoding of stochastic behavior for fine-grained variability [Guo et al., 2024, Gu et al., 2025]. These properties make codified profiles ideal for faithful, efficient, and adaptive role-playing, especially with smaller models.

To evaluate the effectiveness of codified profiles, we construct a new Fandom Benchmark comprising 83 characters and 5,141 scenes curated from Fandom, spanning mangas, novels, and television series. Each scene is paired with ground-truth character actions extracted from original narrative artifacts. To assess alignment between model outputs and expected behavior, we employ a validated LLM-based natural language inference (NLI) scoring framework [Bowman et al., 2015], which measures logical entailment between generated responses and reference actions.

Our experiments start with a comparison between our codified profiles and traditional textual profiles, and then dive deep into the effect of evolving profiles that update sequentially along the episode timeline, the influence of profile-driven randomness, and the model's Best@K performance under stochastic response settings. Across all settings, codified profiles significantly improve behavioral consistency, adaptability, and diversity, especially when used with smaller LLMs.

We further investigate codification strategies, focusing on the granularity of profile segmentation. Our findings indicate that paragraph-level segments strike the best balance between fidelity and computational efficiency: larger segments often obscure fine-grained logic, while smaller ones lead

to excessive fragmentation and redundant condition checking. We also present case studies that demonstrate the concrete advantages of codified profiles over prompt-based interpretations. Our key contributions are as follows:

- We propose codified profiles as executable and interpretable representations of character logic that provide persistence, updatability, and controllable randomness beyond the limitations of prompt-based role-playing.
- We introduce a new benchmark, constructed from Fandom-sourced scenes and characters, for evaluating role-playing consistency. We plan to open-source this dataset to facilitate future research.
- We present an in-depth analysis of codified profiles, including distilling condition checkers, profile segmentation trade-offs, and demonstrate their impact through detailed case studies.

## 2    Background

The rise of LLMs has heightened users' expectations not only for accurate answers but also for emotionally supportive interactions [Chen et al., 2024a,b]. Role-playing has emerged as a trending task to meet this demand, enabling engaging interactions such as simulating fictional characters [You et al., 2022, Thoppilan et al., 2022]. It allows LLMs to adopt varied profiles, supporting storytelling [Brahman et al., 2021, Qin et al., 2024], games [Yu et al., 2025], and metaverse applications [Zhou, 2023, Lifelo et al., 2024]. We present a comprehensive overview of role-playing developments, including training, inference, grounding, evaluation, and the evolving role of codified profiles in both traditional games and modern LLMs.

**Training and Inference**    The most straightforward way to build a character into an LLM is to train on how the character reacts to different conditions. Such training data can be obtained from existing plots [Li et al., 2023] or profile-based data synthesis [Shao et al., 2023]. However, the scarcity of plots (especially for original characters) and insufficient grounding for synthesis [Peng and Shang, 2024] cause hallucination challenges [Sadeq et al., 2024] caused by fine-tuning. During the inference, LLMs are generally prompted with statements from profiles to mitigate hallucinations, which requires a strong grounding system.

**Grounding and Updating**    The grounding system in role-playing models what and how to append profile information to prompts. The "how" part is modeled similarly to traditional retrieval-augmented generation (RAG) [Lewis et al., 2020] using similarity metrics to retrieve the most relevant statements [Karpukhin et al., 2020, Liu et al., 2023, Salemi et al., 2024]. The "what" part discusses how to preprocess the statements for retrieval, including formatting and segmentation [Yuan et al., 2024]. Advanced role-playing systems also expect statements to be dynamically updated during the interaction [Zhou et al., 2024] for both working memory and long-term profile. Such updates generally "backpropagate" self-reflection [Shinn et al., 2023] to the profile, which refines the original statements in a textual gradient style [Yuksekgonul et al., 2025].

**Evaluation**    There are evaluations for both understanding and generation in role-playing. For understanding, factuality checking and multiple-choice question answering benchmarks [Salemi et al., 2024, Yuan et al., 2024] are developed to evaluate the understanding of characters. For generation, the reference comes from ground-truth scenes [Wang et al., 2023b] and human judgment [Chen et al., 2023, 2025]. The profile is an essential reference for human evaluators, especially when evaluators are unfamiliar with the character. The evaluation can also be decomposed into simple tasks like natural language inference [Bowman et al., 2015] that can be trustfully assigned to neural models for automated evaluation [Peng and Shang, 2024]. Some works also apply LLMs to evaluate more advanced attributes such as value and personality [Shao et al., 2023, Wang et al., 2024].

**Code in Role-playing**    Building characters with code is strongly connected to the development of video games [Stanley et al., 2006, Gallotta et al., 2024]. The most common practice is to hardcode the actions of non-player characters (NPCs), making them actors in prewritten scripts. They are only allowed to intellectually interact with players in some subscenes (e.g., battle, chess), which hardly affect the storyline progression. In role-playing, codes and LLMs represent two extremes in interaction freedom, which are synergized in our codified profile framework to reach a balanced point by generating and executing prewritten symbolic logic by a flexible neural system.

## 3 Codified Profile

### 3.1 Role-playing Preliminary

A pure role-playing system models character logic as "Input a scene $s$, output a response $r$ to the scene as character X", where the response can be utterances, actions, or both. Here we use $\text{LLM}(\cdot)$ to represent the role-playing systems as our discussion scope will be LLM-driven role-playing. The LLM is prompted to role-play as characters with a system prompt $I_{\text{role-play}}$, which does not contain information of any character to be universally applied for role-playing: $r = \text{LLM}(s \mid I_{\text{role-play}})$.

Textual profile $P = [p_1, p_2, \cdots, p_N]$ ($p_i$ represents segments of $P$ like paragraphs) further grounds the role-playing with in-context information about character logic. With $P$ appended to the input, the role-playing LLM can output its response $r$ directly or after reasoning with chain-of-thoughts (CoT) [Wei et al., 2022] $cot$ prompted from CoT reasoning instruction $I_{\text{CoT}}$.

$$r = \text{LLM}(s \mid P, cot, I_{\text{role-play}}, I_{\text{CoT}}) \text{ where } cot = \text{LLM}(s \mid P, I_{\text{role-play}}, I_{\text{CoT}}) \tag{1}$$

### 3.2 Codifying Profiles

Our motivation to codify profiles emerges from the instability for LLMs to interpret textual profiles during inference (the CoT stage $cot = \text{LLM}(s \mid I_{\text{role-play}}, I_{\text{CoT}}, P)$), which completely relies on LLM's reasoning accuracy without missing or misinterpretation. This reliance creates a brittle dependency on the model's reasoning over long, natural language inputs, often leading to errors in complex scenes. Codification mitigates this fragility by offloading as much reasoning as possible into deterministic control logic, allowing the LLM to focus on localized decisions such as condition evaluation and action selection. All prompts and templates for implementation are in Appendix B.

In practice, we utilize LLM's coding ability to codify each profile segment $p_i$ into an executable function $f_i : s \to p_i^{\text{trig}}$ (Code implementation: `parse_by_scene(scene)` $\to$ `triggered_statements`), which returns the possible reactions $p_i^{\text{trig}}$ in the scene $s$ based on the logic written in $p_i$. LLM then combines all $p_1^{\text{trig}}$ to summarize a final response $r$.

$$r = \text{LLM}(s \mid [f_1(s), f_2(s), \cdots, f_N(s)], I_{\text{role-play}}) \tag{2}$$

To enhance flexibility while preserving structure, each function $f_i$ can incorporate semantic checks using a callable helper function `check_condition(scene, question)`. This callable allows $f_i$ to evaluate context-sensitive conditions that are difficult to express through static rules alone. For example, a rule like "Whenever being insulted, X will become outrageous" may be codified as:

```
if check_condition(scene, "Is X being insulted?"):
    triggered_statements.append("X become outrageous")
```

Internally, `check_condition` queries the role-playing LLM with a natural language question and interprets the logits over verbalizers (e.g., *"yes"*, *"no"*, *"unknown"*) to yield boolean or uncertain judgments. This enables codified profiles to combine deterministic control flow with semantic nuance, supporting persistent and context-aware behavior across scenes. As shown in Figure 15 (Appendix F), this approach covers personality, relationships, and superpowers. By shifting the task from profile-based reasoning to localized condition classification, codification allows smaller LLMs to perform competitively, making it well-suited for low-resource deployments.

### 3.3 Evolving by Storyline

An essential advantage of codified profiles is long-term maintenance, considering the Github repository updates according to discovered issues. Codified profiles can also be updated along with the storyline, and each profile version reflects the character logic in different episodes. We plot the workflow to evolve codified profiles in Figure 2. Issues in role-playing emerge from predicted response $r$ mismatching with ground-truth response $r_{\text{ref}}$ based on scene $s$. We select natural language inference (NLI) as the initial signal for profile updating, which discriminates $r$ as {*"contradicted"*, *"neutral"*, *"entailed"*} based on $r_{\text{ref}}$.

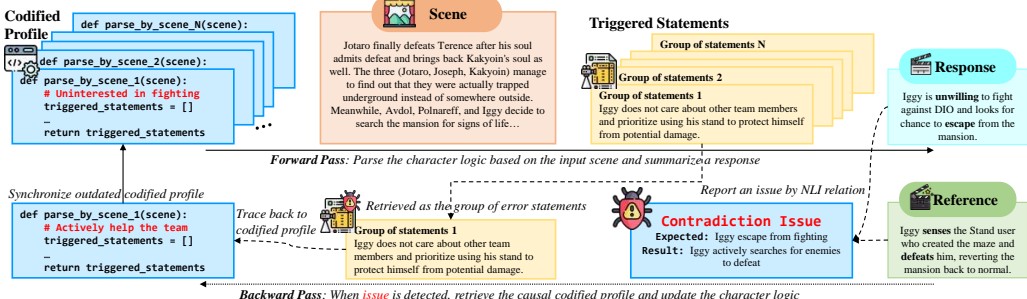

Figure 2: The evolving mechanism of codified profile to synchronize character with the storyline.

When $\text{NLI}(r_{\text{ref}}, r) \in \{$ *"contradicted"*, *"neutral"*$\}$, the issue will be reported to the response generation stage, citing *"contradicted statement"* (contradicted) or *"relevant but not detailed statement"* (neutral). Then, LLM is prompted to detect the group of `triggered_statements` that fits such accusation. Supposing $p_j^{\text{trig}}$ as one of the proposed `triggered_statements` groups, $f_j$ will be revised as a problematic character logic. Finally, $f_j$ will be updated based on the input scene $s$ together with $p_j^{\text{trig}}, r_{\text{ref}}, r$ to synchronize the character logic with the storyline.

Textual profiles are not inherently unevolvable. If formatted into logic like `if-then-else`, they can be incrementally revised. However, most textual profiles are not logically formalized. Moreover, updating texts mimics code debugging, yet lacks the precision or modularity of code debugging.

## 3.4 Controllable Randomness Simulation

LLMs struggle to simulate precise randomness, as generation is governed by temperature rather than probability in instructions. For example, when asked to respond with equal probability between positive and negative sentiment, they often show biased, unbalanced behavior due to training data and decoding preferences [Guo et al., 2024, Gu et al., 2025].

Codified profiles overcome this limitation by externalizing randomness into explicit Python control logic, using constructs like `random.choice([...])` or `random.random() < p`. Instead of relying on the LLM's generation-time sampling, this approach enforces precise stochastic behavior directly within the code. For example, to simulate random sentiment:

```
sentiment_statement = random.choice(["X is positive.", "X is negative."])
```

By embedding such logic in codified profiles, we achieve fine-grained control over behavioral randomness, ensuring both reproducibility and tunability. This approach is particularly advantageous in applications where controlled variability is essential, such as user interaction.

## 4 Fandom Benchmark

Although various high-quality benchmarks [Wang et al., 2024, 2025] exist for evaluating role-playing in LLMs, they tend to emphasize dialogue interactions over the complicated execution of situational actions in grounded contexts. Moreover, many benchmarks rely on data synthesized by LLMs or sparsely annotated scripts [Wang et al., 2023b, Chen et al., 2024a], which restricts them in evaluating an in-depth understanding of massive character logic.

We propose using Fandom[4] as a rich source of character profiles and structured story summaries to build a behavior-centric role-playing benchmark. For each narrative segment, we prompt `gpt-4.1` to extract character actions (e.g., Green-highlighted parts in the storyline of Figure 3) and treat preceding text as the scene. Each scene-action pair is further augmented with a carefully crafted guiding question (by `gpt-4.1`), which constrains responses to be reference-relevant without revealing answers. During evaluation, the role-playing LLM receives the character profile (spoiler-free, cleaned by `gpt-4.1`), scene, and question, and predicts the character's next action. Predictions are scored using an LLM-based NLI system: entailed predictions score 100, neutral 50, and contradictions 0.

---

[4]https://www.fandom.com/

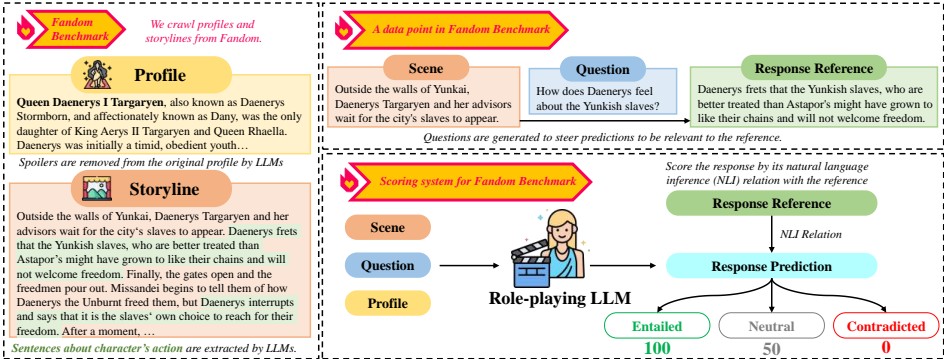

Figure 3: Curation and evaluation scenarios of our Fandom Benchmark.

We select 6 reputational artifacts to cover various themes of stories: *"Suzumiya Haruhi (Haruhi)"*, *"K-On!"*, *"JOJO's Bizzare Adventure (JOJO)" (Season 3)*, *"Fullmetal Alchemist (FMA)" (2009)*, *"A Game of Thrones (AGOT)" (Season 1-3)*, and *"Avatar: The Last Airbender (ATLA)" (Book1)*. The benchmark covers diverse artifacts from sci-fi to adventure across manga, novels, and TV series, featuring 83 characters in 5,141 scenes. Appendix G provides detailed statistics, character information, storylines, and example profiles for clarity.

## 5 Experiment

### 5.1 Evaluation Scenarios

For evaluation, we use `gpt-4.1` as the codification LLM to convert natural language character profiles into codified profiles.[5] Then, we employ `llama-3.1-8b-instruct` as the role-playing LLM. This model is chosen to ensure experimental efficiency, including tractable forward pass counts and full access to output logits for

| Setup | Update Profile | Import Random | Sampling | Metric |
|-------|---------|---------|----------|--------|
| Basic | ✗ | ✗ | ✗ | NLI |
| Evolving | ✓ | ✗ | ✗ | NLI |
| Stochastic | ✗ | ✓ | ✓ | Best@K (NLI) |

Table 1: Detailed setups in experiments.

analysis.[6] For each scene in our Fandom benchmark, the role-playing LLM is prompted with a given profile and is asked to produce the character's next action. This response is scored using LLM-based (`gpt-4.1`; We manually check its 5 NLI scoring cases for all 83 characters, and find 92.35% entailed, 90.91% neutral, and 88.89% contradicted judgments align with human evaluation, which validates the usage of LLM for automated NLI scoring) NLI against the ground-truth, as mentioned in the benchmark section. We further extend the basic scenario to **evolving profile** and **stochastic response**.

In **Evolving Profile**, scenes are evaluated sequentially; after testing on the $i$-th scene, if the predicted action is not entailed (based on NLI), the profile is updated to reflect revised behavioral logic before testing scene $(i+1)$, as detailed in § 3.3. In practice, we update one function for each reported issue for efficiency. In **Stochastic Response**, the role-playing LLM generates $K$ responses per scene using a fixed codified profile with probabilistic constructs (e.g., `random.choice`) and temperature $< 0.7$. The codification LLM is explicitly prompted to introduce randomness into control flow. We evaluate using Best@K, which assigns the final score for each scene based on the highest NLI score among the $K$ sampled responses, capturing the model's potential best alignment. Difference between setups can be referred to Table 1. Before the experiments, we check the codification ability of `gpt-4.1`, which shows competent performance as shown in Appendix E. All prompts are included in Appendix B.

### 5.2 Baselines

- **Vanilla** The role-playing LLM is prompted with only the scene. This setting serves as a lower bound, representing performance with parameterized knowledge when profile data is unavailable.
- **Textual Profile** The full natural language character profile is provided alongside the scene. This is the standard grounding method in most existing role-playing systems.

---

[5]In Appendix 5.6, we discuss the influence of using different codification LLMs.

[6]GPT-4 series are also found memorizing artifacts like AGOT [Chang et al., 2023], injecting evaluation bias.

| Artifact | | Haruhi | K-On! | JOJO | FMA | AGOT | ATLA | Average |
|---|---|---|---|---|---|---|---|---|
| **Main** | #Character | 5 | 5 | 7 | 5 | 11 | 4 | 5.3 |
| | Vanilla (No Profile) | 63.53 | 63.60 | 59.70 | 67.10 | 64.93 | 66.40 | 64.21 |
| | Textual Profile | 70.53 | 68.47 | 60.64 | 68.02 | 61.52 | 66.70 | 65.98 |
| | Codified RAG | 69.03 | 66.81 | 59.52 | 69.14 | 65.39 | 66.33 | 66.04 |
| | Codified Profile | 72.14 | 69.23 | 63.92 | 69.60 | 67.51 | 67.90 | **68.38** |
| | + Textual Profile | 72.81 | 71.17 | 63.42 | 71.36 | 65.60 | 67.37 | **68.62** |
| **Minor** | #Character | | 4 | 9 | 7 | 19 | 7 | 9.2 |
| | Vanilla (No Profile) | | 65.12 | 62.53 | 61.17 | 67.65 | 66.89 | 64.67 |
| | Textual Profile | | 65.88 | 64.70 | 65.83 | 66.21 | 65.88 | 65.70 |
| | Codified RAG | N/A | 63.28 | 68.13 | 66.97 | 68.62 | 66.65 | 66.73 |
| | Codified Profile | | 68.59 | 70.06 | 68.21 | 70.51 | 71.95 | **69.87** |
| | + Textual Profile | | 67.97 | 68.94 | 69.22 | 71.83 | 71.86 | **69.96** |

Table 2: Role-playing performance based on different profiles. N/A: No minor character in Haruhi.

- **Codified RAG** A simple codification strategy that implements retrieval-augmented generation (RAG) over the profile. Each segment of the profile is wrapped in a `check_condition` function to determine its relevance to the current scene (`if check_condition(scene, question): triggered_statements.append(segment)`). This method evaluates whether logic-based codification offers deeper character understanding than shallow relevance matching.
- **Reasoning Mechanism** We incorporate chain-of-thoughts prompting before the action prediction step. To fairly evaluate reasoning efficiency, we apply CoT both on top of the Textual Profile and the Codified Profile, and vary the average reasoning chain length from 0 upward. We report performance and forward pass count to assess the trade-off between response quality and computational cost.

## 5.3 Role-playing Results

In Table 3 and Figure 4, we present the performance of various role-playing methods. Results show that all grounding-based approaches outperform the vanilla (No Profile) baseline, underscoring the importance of contextual character guidance. The result validates our core claim that enforcing consistent character logic yields better outcomes than relying on the LLM's implicit reasoning, as codified profiles outperform Textual Profiles across both main and minor characters. Codified profiles also surpass Codified RAG, demonstrating that their advantage lies not just in retrieving relevant content but in expressing structured, executable behavior logic for more persistent and faithful character simulation.

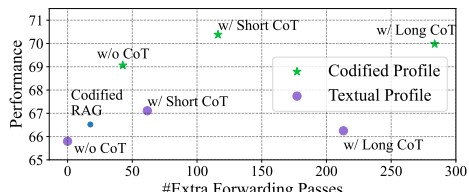

Figure 4: The role-playing performance with reasoning mechanism.

Adding textual profiles on top of codified profiles provides further but marginal improvements, especially when the performance difference between code and text is small. This suggests that textual prompts can sometimes complement information that may have been simplified or abstracted during codification, enhancing the expressiveness of the final behavior. In Appendix C, we continue the discussion with human/LLM preference and evaluation on original (out-of-script) scenes.

**Reasoning** Figure 4 compares codified reasoning with chain-of-thoughts reasoning, averaged over all characters. The results show that the codified reasoning flow achieves strong performance with fewer reasoning steps than chain-of-thoughts, demonstrating it as a more efficient grounding mechanism compared to approaches that rely heavily on runtime reasoning. Notably, performance improves further when codified profiles are combined with targeted reasoning, indicating that our method can benefit from additional inference when applied judiciously. In contrast, the long chain-of-thought (CoT) baseline underperforms, and manual inspection reveals that it often suffers from overthinking, introducing unnecessary complications that derail character-consistent behavior.

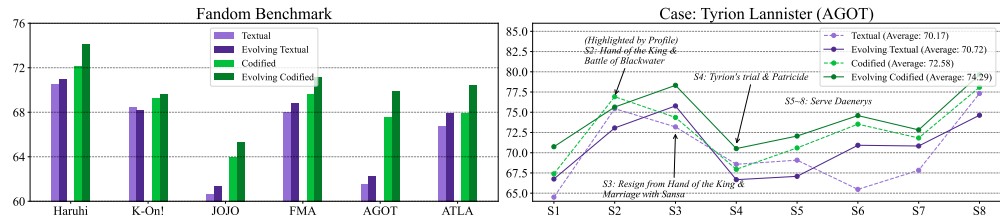

Figure 5: Role-playing performance with evolving profiles. **Left:** Performance gain from evolving. **Right:** Case of *"Tyrion Lannister"* in AGOT to analyze the working mechanism of evolving.

**Evolving**    In Figure 5, we examine the effect of evolving codified profiles by the storyline on all characters inside the Fandom benchmark. Evolving for textual profiles follows a similar pipeline as the codified profile by select-then-revise. The results show codified to be more efficient than textual profiles in evolving, as it directly models character logic. The evolving mechanism works better on adventure series rather than relaxing anime like K-On! as they involve more character growth.

The right subfigure presents a case study on *"Tyrion Lannister"*, evolving along with all 8 seasons, illustrating how evolving profiles maintain alignment with narrative shifts. Tyrion's profile is well-grounded in Season 2 during the Battle of Blackwater, his defining moment centered by the original profile. However, Seasons 3~4 show a heightened need for evolution, surpassing even later seasons when he joins Daenerys, reflecting rapid character development post-battle. This highlights that the most critical synchronization occurs after major turning points. Codified profiles with evolution enable timely updates to capture these shifts, preserving logic-storyline consistency.

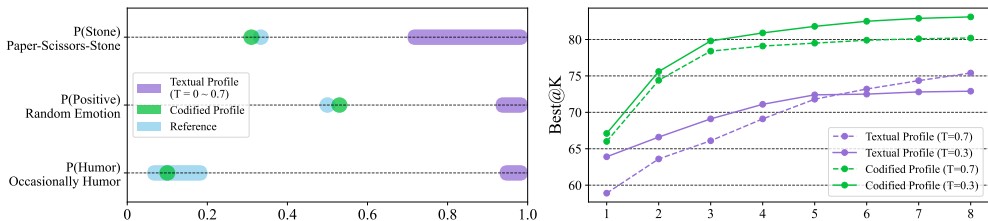

Figure 6: **Left:** Cases on controllable randomness simulation by textual and codified profiles. **Right:** Scenario coverage comparison, evaluated by Best@K performance with stochastic responses.

Figure 6 compares randomness control in textual vs. codified profiles across three simulation cases (by 100 runs): choosing paper-scissor-stone (with equal probability), responding with positive or negative emotions (with equal probability), and occasionally adding humor (considering $5\% \sim 20\%$ as acceptable probability for *"occasionally"*). Textual prompting (Temperature 0~0.7) shows significant bias, overproducing certain responses, while codified profiles accurately follow specified probabilities via explicit control flow. Appendix F details the codified logic in Figure 17, which aligns with intended behavior.

The right subfigure shows the application of randomness simulation to complex characters in the Fandom benchmark. As expected, codified profiles implemented via explicit randomness in code control flow achieve stronger Best@K performance, meaning they explore more reasonable reactions. Textual profiles rely on raising temperature to explore variability, which degrades single-run precision and introduces inconsistency. By decoupling behavioral exploration from sampling noise, codified profiles deliver diversity and accuracy at low temperatures, making them diverse yet high-quality for role-playing. We further showcase how randomness enriches characters in Figure 18.

## 5.4    Smaller Role-playing LLMs

We compare vanilla prompting, textual profiles with chain-of-thought, and codified profiles across LLaMA-3 models of 1B (3.2), 3B (3.2), and 8B (3.1) parameters in Figure 7, and observe that the advantage of codified profiles grows as model size decreases. Notably, smaller LLMs with codified profiles rival the larger LLMs with textual profiles for both 1B and 3B cases. Manual inspection shows smaller

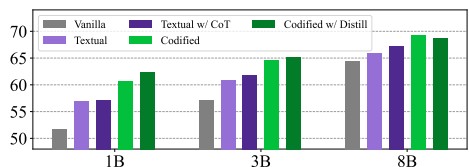

Figure 7: The role-playing performance with smaller role-playing LLMs.

| Artifact | | Haruhi | K-On! | JOJO | FMA | AGOT | ATLA | Average |
|---|---|---|---|---|---|---|---|---|
| Main | `kimi-k2` | 74.81 | 67.71 | 62.70 | 68.89 | 67.32 | 64.62 | 67.68 |
| | `qwen3-instruct` | 76.24 | 67.89 | 63.80 | 70.27 | 67.46 | 64.23 | 68.32 |
| | `qwen3-coder` | 74.25 | 69.82 | 63.62 | 70.62 | 67.88 | 69.94 | 69.36 |
| | `gpt-4.1-mini` | 73.33 | 68.58 | 62.71 | 68.07 | 67.03 | 68.54 | 68.04 |
| | `gpt-4.1` | 72.14 | 69.23 | 63.92 | 69.60 | 67.51 | 67.90 | 68.38 |
| | `claude-3.7-sonet` | 76.86 | 68.53 | 63.35 | 70.82 | 68.69 | 68.73 | 69.50 |
| | `claude-4.0-sonet` | 75.07 | 68.99 | 63.85 | 74.26 | 67.22 | 70.82 | **70.04** |
| Minor | `kimi-k2` | N/A | 67.70 | 65.79 | 65.53 | 71.70 | 64.09 | 66.96 |
| | `qwen3-instruct` | | 69.48 | 66.08 | 65.64 | 71.71 | 65.70 | 67.72 |
| | `qwen3-coder` | | 71.63 | 67.51 | 64.44 | 73.64 | 72.06 | 69.86 |
| | `gpt-4.1-mini` | | 71.65 | 64.66 | 70.35 | 69.77 | 70.43 | 69.37 |
| | `gpt-4.1` | | 68.59 | 70.06 | 68.21 | 70.51 | 71.95 | 69.87 |
| | `claude-3.7-sonet` | | 75.05 | 67.05 | 67.04 | 73.79 | 69.25 | 70.44 |
| | `claude-4.0-sonet` | | 71.79 | 67.66 | 70.94 | 74.63 | 73.72 | **71.75** |

Table 3: Role-playing performance based on codification LLMs with the 8B role-playing LLM.

models often fail to reconstruct character logic from
free-form text but still perform well when logic is preprocessed by stronger LLMs (e.g. `gpt-4.1`)
into explicit conditions. This supports our central claim in the introduction: codified profiles reduce
reliance on the LLM's implicit and inconsistent reasoning by enforcing structured, interpretable
behavior, which is especially critical for smaller models with limited reasoning ability.

**Distilled Condition Checker**   As `check_condition(scene, question)` is a discriminative
task, we explore the use of a distilled classifier to improve both performance and efficiency. Instead of
querying the role-playing model for every condition, we distill from `gpt-4.1`'s condition-checking
outputs using 415 scenes (5 per character, $8\%$ of all scenes) and obtain 20,759 labeled discrimination
cases. A 3-class `deberta-v3-base` model (0.1B) [He et al., 2021] is trained on 90% of the data for
5 epochs and achieves 70.53% consistency with `gpt-4.1` on the held-out 10%. As shown in Figure 7,
this design enables a highly efficient 1B + Code + Distill system that approaches the performance
of 8B + Text, demonstrating the benefit of codified reasoning to smaller LLMs. On a larger 8B
role-playing LLM, the distilled classifier maintains a comparable performance with significantly
better efficiency, as illustrated in the efficiency analysis in Appendix D. We have also checked the
performance excluding the $8\%$ data for distillation, which reaches a consistent conclusion, supporting
the generalization of the distilled condition checker.

## 5.5   Segmentation Strategy

Figure 8 compares section, paragraph, and sentence-
level codification. Finer segmentation improves role-
playing, with a notable gain from section to para-
graph. However, the marginal benefit from paragraph
to sentence comes at a high computational cost, al-
most tripling forward passes. Paragraphs offer a natu-
ral grouping of related statements, balancing fidelity
and efficiency. Thus, we adopt paragraph-level seg-
mentation for the main experiments.

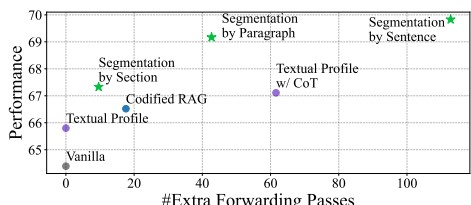

Figure 8: The role-playing performance with
different segmentation strategies.

## 5.6   Codification LLM Comparison

Table 3 compares role-playing performance across different codification LLMs, showing that overall
effectiveness generally correlates with the model's coding ability on the public leaderboard. However,
performance varies by artifact, suggesting that codifying character profiles requires more than
syntactic code generation, demanding a nuanced understanding of different personalities, contexts,
and behavioral logic that vary among artifacts. Notably, `claude-3.7-sonet` achieves the highest
average, indicating stronger alignment in both structural and semantic aspects of profile codification.

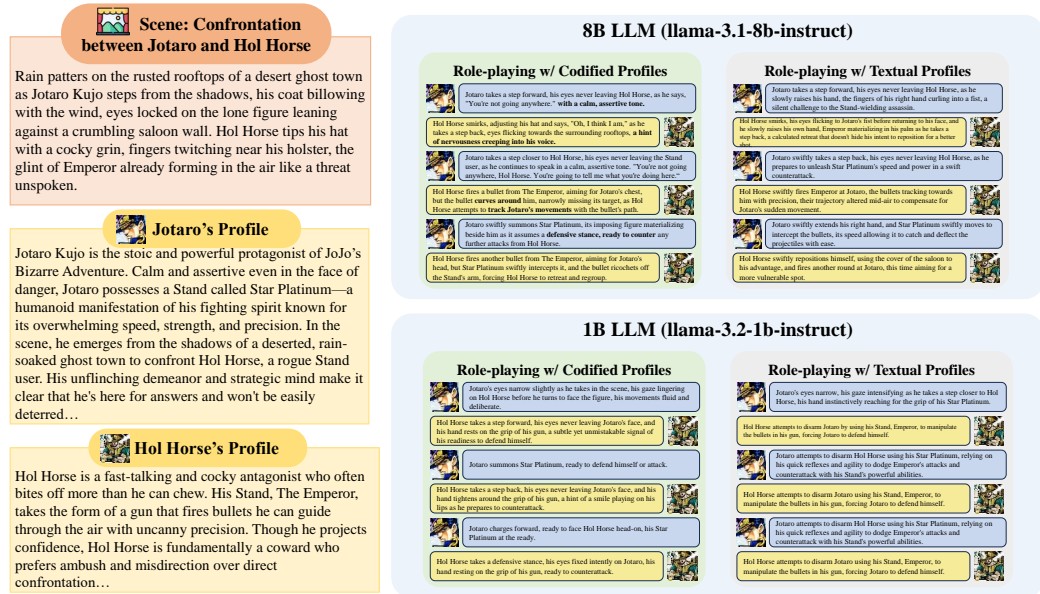

Figure 9: Case study of multi-turn role-playing in a given scene: *"Jotaro confronting Hol Horse."* *(JOJO's Bizarre Adventure)*

# 6 Case Study

Figure 9 illustrates a case study that highlights the advantage of codified profiles in a direct role-play scenario. We construct a dynamic scene featuring a confrontation between *"Jotaro"* and *"Hol Horse"* from JOJO, and prompt LLMs to role-play as each character by *"Propose the next action that {character} takes in response."* Both textual and codified profiles are evaluated using 8B and 1B models to compare performance across model scales and profile formats.

Codified profiles enhance role-playing coherence and character fidelity by embedding consistent personality traits and logically grounded actions into explicit control flow. For instance, when Hol Horse fires a bullet and misses, the codified logic triggers Jotaro to summon Star Platinum in immediate defense, mirroring canonical behavior. In contrast, textual profiles often produce generic and less impactful responses, such as Jotaro merely raising a fist and stepping back without clear narrative consequence. By offloading complex reasoning into structured logic, codified profiles allow even smaller LLMs (e.g., 1B models) to engage in reactive exchanges, avoiding repetitive outputs like *"Hol Horse attempts to disarm Jotaro."* This explicit structure reduces reliance on the model's implicit reasoning and enables believable, narratively aligned interactions that would otherwise require far greater model capacity. We further include extended case studies for codified profile workflow in Appendix F, which can be referred to understand the mechanism and advantage of codified profiles.

# 7 Conclusion

We propose codified profiles, which offer a principled framework for character role-playing by compiling behavior into interpretable, executable functions. Our experiments demonstrate that this approach outperforms prompt-based methods in persistency, updatability, and controllable randomness, while enabling strong performance even in small models. These findings highlight codified profiles as a practical foundation for building efficient, locally deployable role-play agents across a wide range of applications. Future work, as discussed in Appendix A, includes extending codification with richer operators and helper tools, modeling environmental logic for world-level role-playing, and exploring the scope of codifiable textual content.

## Acknowledgement

This work aims to contribute not only to the research community but also to a broader ACG community by introducing more powerful role-playing agents. It is also done in memory of the 17th *Koishi's Day* (May 14th), 2025, since the release of TH11, Touhou Chireiden ∼ Subterranean Animism[7] in 2008.

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

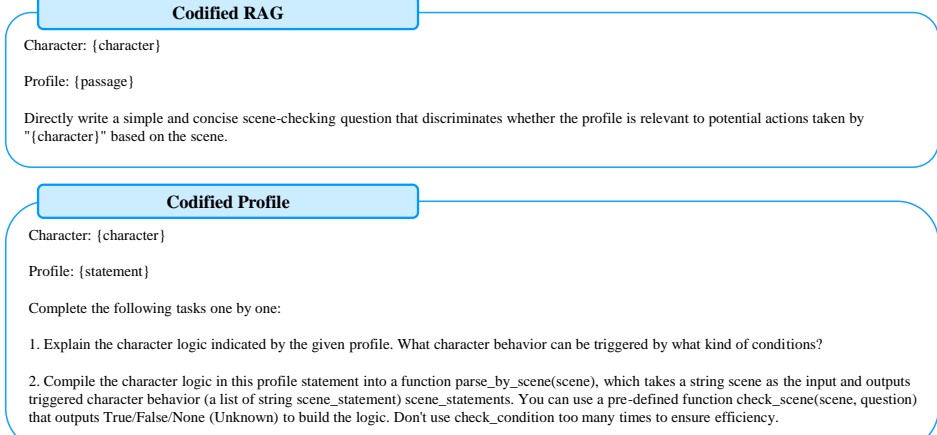

Figure 10: The preprocessing prompts used in our experiments.

# A  Limitation and Future Work

As a pioneering study that integrates executable code into character-based role-playing, this paper focuses on presenting the broad applicability and core benefits of codified profiles. However, several promising further explorations are remained for future works.

**More operators and helper functions**  First, while our framework centers on the `check_condition` helper function due to its semantic clarity and general applicability, future work can introduce additional operators and helper tools to support more complex behaviors. For example, in role-playing game (RPG) scenarios, codified profiles could include arithmetic operations for managing health or damage, procedures for activating skills, or functions for spatial movement and positioning. These extensions would enable more dynamic and interactive role-play beyond conditional responses.

**World-level role-playing**  Second, codification can be expanded beyond individual characters to represent the fictional world itself. Codifying environmental rules, societal structures, or magical systems would allow models to simulate consistent world behavior and responses. While this paper does not pursue world-level codification due to the lack of benchmarks for evaluating environmental reactivity, it presents a promising direction for future research in narrative simulation.

**Scope of codification**  Finally, codification introduces a potential risk of information loss by abstracting nuanced textual descriptions into structured logic. We view this as a trade-off that can be managed: codified functions can always reference the original profile when needed, and conservative codification that only translates explicit control flows already improves performance. Developing hybrid strategies that balance interpretability and fidelity remains an important direction for making codified profiles more robust and expressive.

These limitations reflect deliberate design choices to focus on a broad and extensible framework. We hope they inspire future work that builds on codified reasoning to support richer, more controllable, and scalable role-playing systems.

# B  Prompts and Templates

Figures from 10 to 13 show the prompts used in our experiments for result reproduction.

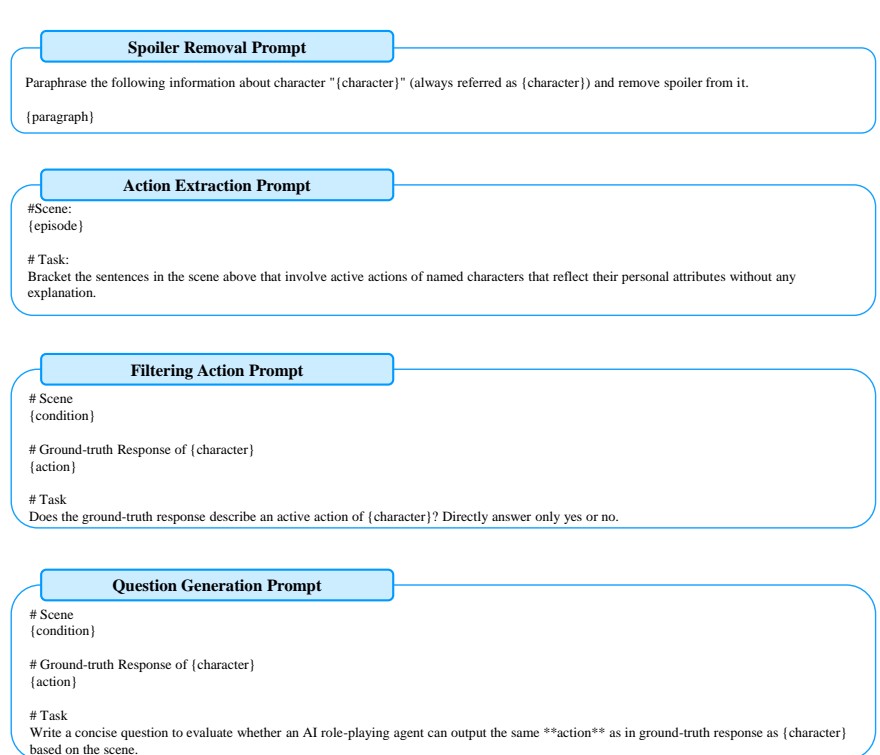

**Spoiler Removal Prompt**

Paraphrase the following information about character "{character}" (always referred as {character}) and remove spoiler from it.

{paragraph}

**Action Extraction Prompt**

#Scene:
{episode}

# Task:
Bracket the sentences in the scene above that involve active actions of named characters that reflect their personal attributes without any explanation.

**Filtering Action Prompt**

# Scene
{condition}

# Ground-truth Response of {character}
{action}

# Task
Does the ground-truth response describe an active action of {character}? Directly answer only yes or no.

**Question Generation Prompt**

# Scene
{condition}

# Ground-truth Response of {character}
{action}

# Task
Write a concise question to evaluate whether an AI role-playing agent can output the same **action** as in ground-truth response as {character} based on the scene.

Figure 11: The prompts used for building the Fandom Benchmark.

**Chain-of-Thoughts Prompt**

```
# Background Knowledge
{grounding}
# Scene
{scene}
# Question
{question} Before answering, first think step by step about how to answer the question.
Reasoning:
```

**Issue Detection Prompt**

```
# Scene
{scene}

# {character}'s Predicted Action
{prediction}

# {character}'s Ground-truth Action
{action}

{profile}

# Detected Issue
{issue}

# Task
Detect **one and only one** information block that causes the issue. The output format is as follows:
```json
{
    "reasoning": "...",
    "id": int
}
```
```

**Revision Prompt**

```
The following is the codified profile that causes the issue, revise it slightly to address the detected issue:
(Pre-defined supportive function `check_scene(scene, question)` outputs True/False/None (Unknown) to build the logic.)

```python
{code}
```
```

**Codified Profile w/ Random**

```
Character: {character}

Profile: {statement}

Complete the following tasks one by one:

1. Explain the character logic indicated by the given profile. What character behavior can be triggered by what kind of conditions?

2. Compile the character logic in this profile statement into a function parse_by_scene(scene), which takes a string scene as the input and outputs triggered character behavior (a list of string scene_statement) scene_statements. You can use a pre-defined function check_scene(scene, question) that outputs True/False/None (Unknown) to build the logic. When facing randomness, you shall use `random.random() < p` or `random.choice`. Don't use check_condition too many times to ensure efficiency.
```

Figure 12: The shared prompts used for evolving profile and scholastic response.

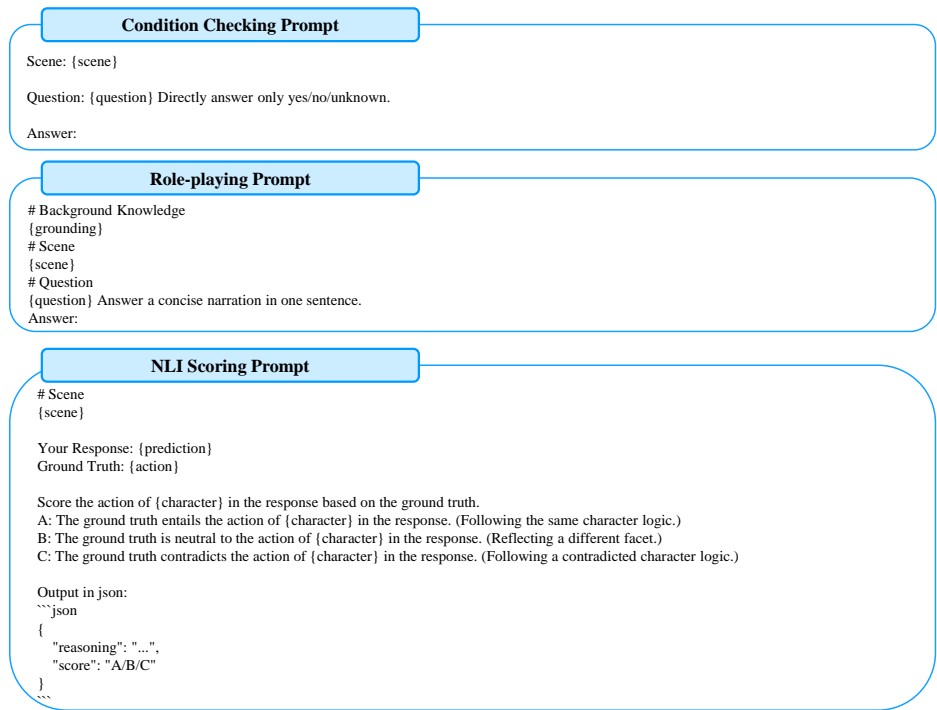

Figure 13: The shared prompts used in our experiments.

## C Human/LLM Preference Evaluation

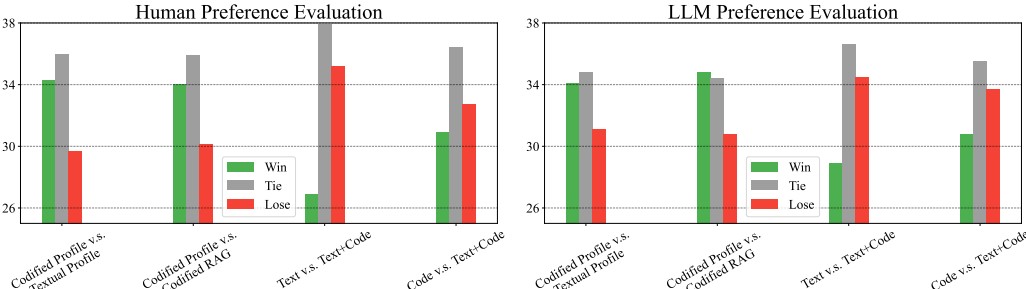

Figure 14: The Human/LLM-based preference evaluation results.

A limitation of the NLI score is its inability to capture fine-grained differences in response quality, since responses with the same NLI relation (e.g., entailed) are assigned the same score. To address this, we conduct a preference-based evaluation where both humans and LLMs are asked to choose the better response between two candidates, given the reference. LLMs are prompted on all scenes with responses shown in both orders to mitigate position bias, and a tie is recorded when the model selects different responses in two runs. Human evaluators assess 5 scenes per character, selecting a preferred response or indicating a tie.

Figure 14 presents the results of this evaluation. First, codified profiles are preferred over both textual profiles and Codified RAG by both human and LLM judges, indicating their advantage in producing precise and contextually faithful responses. While Codified RAG achieves higher NLI scores than textual profiles, it loses the advantage in preference evaluations, likely due to truncation or omission of relevant details during retrieval. Second, ensembling codified and textual responses further improves performance, though codified-only responses contribute more wins, suggesting they provide the stronger baseline. Notably, we observe that LLM-based preferences tend to result in

| Best Performance | | | | Worst Performance | | | |
|---|---|---|---|---|---|---|---|
| Rank | Character | Win Rate | Artifact | Rank | Character | Win Rate | Artifact |
| 1 | Sansa | 81.25 | AGOT | 60 | Winry | 32.00 | FMA |
| 2 | Tywin | 70.00 | AGOT | 59 | Mugi | 34.15 | K-On! |
| 3 | Littlefinger | 68.42 | AGOT | 58 | Riza | 34.78 | FMA |
| 4 | Joffrey | 67.86 | AGOT | 57 | Stannis | 35.00 | AGOT |
| 5 | Jon | 67.57 | AGOT | 56 | Iroh | 38.18 | ATLA |
| 6 | D'Arby | 64.71 | JOJO | 55 | Bran | 38.89 | AGOT |
| 7 | Nagato | 63.64 | Haruhi | 54 | Koizumi | 41.18 | Haruhi |
| 8 | Hol Horse | 62.07 | JOJO | 53 | Sandor | 42.86 | AGOT |
| 9 | Lust | 61.54 | FMA | 52 | Joseph | 42.86 | JOJO |
| 10 | Iggy | 60.71 | JOJO | 51 | DIO | 44.68 | JOJO |

Table 4: Win rates of codified profile against textual profile.

| Model | deberta-v3-base | llama-3.2-1b-it | llama-3.2-3b-it | llama-3.1-8b-it |
|---|---|---|---|---|
| Instance/s | 69.6 | 34.3 | 14.0 | 6.1 |

Table 5: Efficiency comparison among different models as condition checkers.

more ties and occasional misjudgments, likely due to distraction by irrelevant details, reinforcing the importance of human judgment in evaluating nuanced role-play quality.

**Character-wise Comparison**    Table 4 presents a character-level (characters with at least 20 scenes) comparison of win rates between codified and textual profiles (ties are removed), highlighting which characters benefited most from codification. Notably, the top-performing characters such as Tywin, Littlefinger, and D'Arby all exhibit subtle, strategic, or context-sensitive behavior in the source material. These "fickle characters" often require conditional responses based on shifting alliances, hidden motives, or indirect manipulation, making them well-suited to codified logic that explicitly encodes behavioral rules and scenario-based decision-making. In contrast, characters with lower win rates, such as Winry, Mugi, and Riza, tend to exhibit more emotionally driven or straightforward behavior, where textual prompting may suffice and codification provides less added benefit. This pattern suggests that codified profiles are especially advantageous for characters whose actions depend on nuanced reasoning and consistent application of complex behavioral constraints. Such discovery further shows a complementing relation between textual and codified profiles, explaining why the ensemble between them generally works.

**Out-of-script Scenes**    To evaluate the practical applicability of codified profiles beyond scripted narratives, we create 200 interactive scenes featuring diverse scenarios such as playing, battling, and negotiating. Following the methodology of our JOJO case study, we prompt `llama-3.1-8b-instruct` for 3-round interactions using both codified and textual profiles. Codified profiles demonstrate strong generalization, achieving a 35.5% win rate, with 40.0% ties and only 24.5% losses (based on `gpt-4.1` preference), reinforcing their effectiveness in real-world, dynamic role-playing contexts.

# D    Condition Checking Efficiency

In Table D, we present the condition checking efficiency of different models. We calculate efficiency on a single 80G A100 with batch size 1. The distilled `deberta-v3-base` is shown to significantly boost the efficiency when used as the condition checker for LLMs like `llama-3.1-8b-it`.

# E    Codification Performance

We manually analyze the codification ability in the preprocessing stage on 200 cases grouped by different attributes (except for codes with if-depth 4, with only 16 cases). As shown in Table 6, `gpt-4.1` achieves high overall accuracy in translating natural language into structured logic. Precision

| Grouped by | If-depth | | | | w/ Branch | w/ Random | Personality | Ability | Relation |
|---|---|---|---|---|---|---|---|---|---|
| | 1 | 2 | 3 | 4 | | | | | |
| Precision | 98.5 | 96.0 | 92.5 | 93.8 | 98.5 | 91.0 | 97.50 | 92.50 | 96.00 |
| Recall | 94.0 | 93.0 | 89.5 | 81.3 | 95.0 | 96.5 | 96.00 | 94.00 | 98.50 |
| Both | 94.0 | 92.0 | 87.0 | 75.0 | 95.0 | 89.5 | 95.50 | 91.00 | 95.00 |

Table 6: Manual evaluation on codification of profile segments grouped by different attributes.

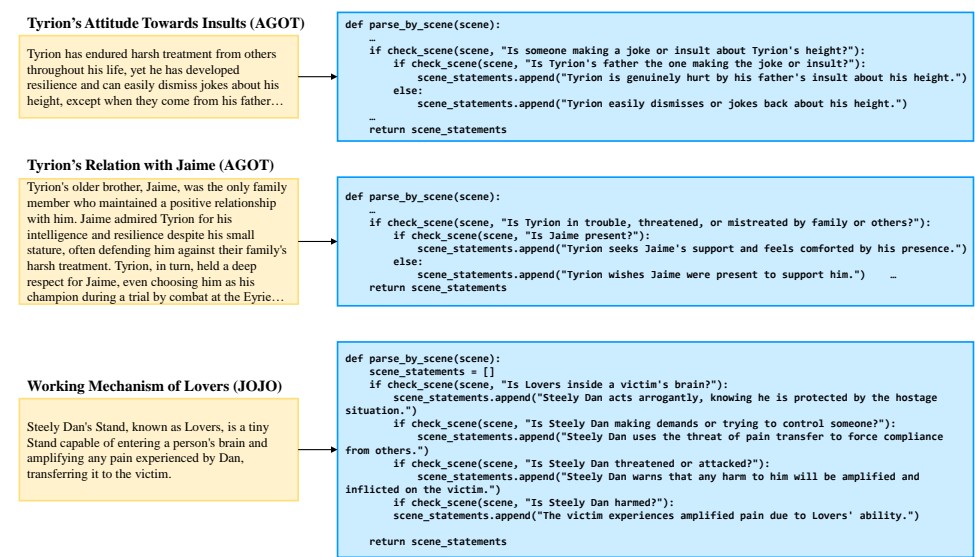

Figure 15: Codification cases for personality, relation, and working mechanism of superpower.

(codified components are all correct) exceeds 98% across shallow control structures and most attributes, indicating strong alignment with intended behavior. Recall (all important information is codified) declines slightly with deeper nesting (81.3% at depth 4), reflecting increased complexity rather than failure. Codification remains robust across personality, ability, and relational traits, with F1 scores above 95% in most categories, demonstrating minimal fidelity loss.

## F  Extended Case Study

**Codification**   In Figure 15, we showcase several codification workflow, including personality, relation and working mechanism of superpower. The cases validate the ability of codified profile to reflect exceptions (*"except when insults come from his father.)"*), situational thoughts (*"Is Jaime present?"*), and superpower checking (*"Dan is harmed"→"The victim is harmed."|"Lovers inside a victim's brain"*). These cases provide a straightforward illustration on how codified profile builds up the character logic.

**Evolving**   An evolving case about Jotaro's fighting strategy is presented in Figure 16, which shows a mismatch between the predicted and reference actions when he is trapped underwater by Dark Blue Moon. The initial codified logic assumed Jotaro always resorts to brute force, resulting in an incorrect prediction where he punches to escape. However, the reference reveals that Jotaro uses the more precise Star Finger technique to blind the enemy and break free. Based on this contradiction, the profile is revised to reflect a more accurate strategy: Jotaro prefers clever abilities like Star Finger when brute strength is ineffective. This update enables the codified profile to better align with his adaptive combat behavior in future scenes.

**Controllable Randomness**   We present the specific codes to demonstrate how codified profiles enable controllable randomness in character behavior. In the first, Robotia plays paper-scissors-rock,

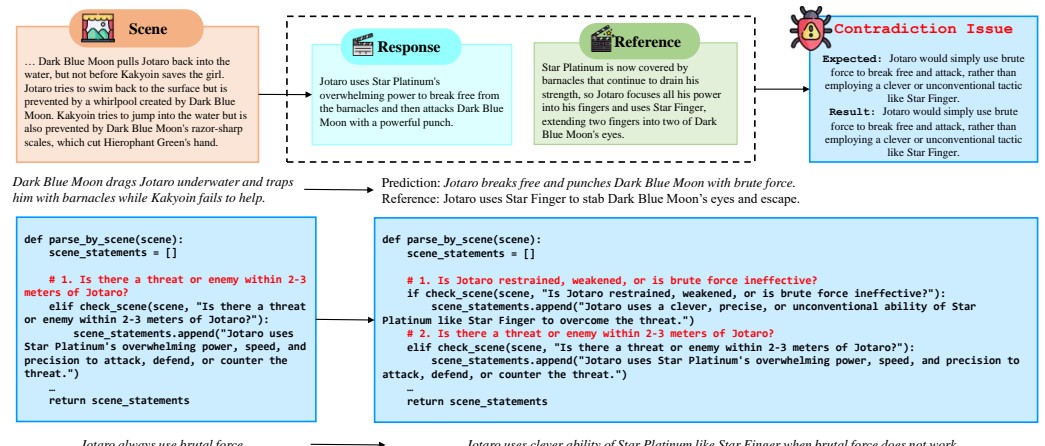

Figure 16: A case for the evolving mechanism for codified profiles.

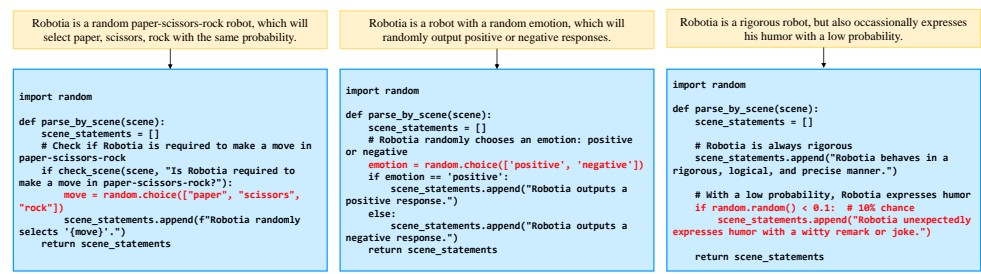

Figure 17: Codification cases with randomness simulation involved.

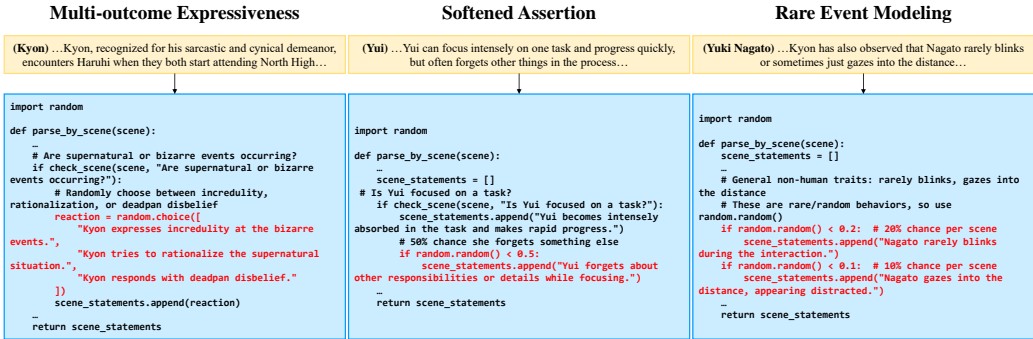

Figure 18: Codification cases of Fandom characters with randomness simulation involved.

| Statistics (Main/Minor) | Haruhi | K-On! | JOJO | FMA | AGOT | ATLA |
|---|---|---|---|---|---|---|
| #Character | 5/- | 5/4 | 7/9 | 5/7 | 11/19 | 4/7 |
| #Scene | 48.4/- | 173.0/49.8 | 111.6/20.8 | 85.8/32.9 | 63.0/23.1 | 214.5/31.1 |
| #Word per profile | 782.6/- | 676.4/449.2 | 925.0/488.8 | 1564.6/1351.1 | 1517.3/1243.5 | 1578.2/740.1 |
| #Paragraph per profile | 11.8/- | 9.2/7.0 | 15.7/9.9 | 16.0/15.1 | 14.6/11.8 | 14.5/9.3 |

Table 7: Important statistics of the Fandom Benchmark.

with each move selected uniformly using `random.choice`, ensuring balanced randomness. In the second, Robotia expresses either a positive or negative emotion with equal probability, highlighting how codified logic maintains behavioral diversity more reliably than prompt-based sampling. In the third, Robotia consistently behaves logically but has a 10% chance of adding humor, showing how low-probability actions can be precisely embedded without disrupting core traits. These examples demonstrate how codified randomness offers fine-grained, interpretable variability beyond what temperature tuning alone can provide.

**Randomness in Real Cases**  Randomness in codified profiles enhances realism by supporting three key concepts: multi-outcome expressiveness, softened assertion, and rare event modeling as shown in Figure 18. For example, Kyon's reactions to bizarre events are randomly selected from a set of consistent yet varied responses, enabling multi-outcome expressiveness. Yui's intense focus includes a 50% chance of forgetting other responsibilities, introducing probabilistic nuance that softens rigid assertions. Nagato's rare behaviors, like not blinking or gazing into the distance, are modeled with low probabilities, capturing subtle traits through rare event modeling. Together, these uses of randomness enrich character behavior while preserving logical structure.

# G  Benchmark and Character Details

- **Haruhi**: *"Suzumiya Haruhi"* is a high school sci-fi comedy centered on a girl who unknowingly possesses godlike powers, dragging her eccentric club into supernatural chaos.
- **K-On!**: *"K-On!"* follows a group of high school girls who form a light music club, bonding over tea, friendship, and their shared love for music.
- **JOJO**: *"JOJO's Bizarre Adventure"* is a multi-generational saga where members of the Joestar family battle supernatural threats with flamboyant powers and style. We select season 3, *"Stardust Crusaders"*, which describes Jotaro's bizarre advantage from Japan to Egypt, fighting against DIO.
- **FMA**: *"Full Metal Alchemist"* tells the story of two brothers who use alchemy in a perilous quest to restore their bodies after a tragic experiment. We select the 2009 edition of FMA.
- **AGOT**: *"A Game of Thrones"* is a gritty political fantasy where noble families vie for power in a brutal world of betrayal, war, and dragons. We focus on the first 3 seasons of the TV series.
- **ATLA**: *"Avatar: The Last Airbender"* is an epic tale of a young boy who must master all four elements to stop a tyrannical war and restore the world's balance. We focus on *"Book One: Water"*.

**Statistics**  Statistics in Table 7 highlight the depth and breadth of the Fandom Benchmark. The average number of scenes for main characters ranges from 48 to 214 scenes each, with minor characters adding dozens more, resulting in a total of 5,141 annotated scenes. Profiles are notably rich, averaging around 1,000 words and 15 paragraphs for main characters, with similarly detailed coverage for minors. This combination of long-form profiles and large-scale scene coverage offers a rigorous testbed for evaluating models' ability to reason over sustained character logic, adapt behavior across contexts, and handle diverse narrative situations with consistency.

**Characters**  In Tables 8 and 9, we present brief introductions of characters involved in our experiments for readers who are unfamiliar with these artifacts. In Figure 19, we also showcase a profile and testing cases for better clarity of our benchmark.

| | | | |
|---|---|---|---|
| Haruhi | main | Haruhi | Haruhi is an eccentric and energetic high school student whose curiosity and unconventional outlook drive the extraordinary events of the "Haruhi" series. |
| | | Kyon | Kyon is a witty and pragmatic high school student who serves as the narrator and reluctant companion to the eccentric Haruhi Suzumiya in the "Haruhi Suzumiya" series. |
| | | Nagato | Nagato Yuki is a quiet, enigmatic member of the SOS Brigade in the "Haruhi Suzumiya" series, known for her extraordinary intelligence and mysterious origins. |
| | | Koizumi | Koizumi is a mysterious and ever-smiling transfer student in the "Haruhi Suzumiya" series, who serves as an esper and a key member of the SOS Brigade. |
| | | Asahina | Asahina Mikuru is a shy and gentle upperclassman in the "Haruhi Suzumiya" series, often roped into the SOS Brigade's antics as their adorable and mysterious "mascot." |
| K-On! | main | Yui | Yui Hirasawa is the cheerful and airheaded lead guitarist of the high school band in the anime "K-On!", known for her infectious enthusiasm and love of sweets. |
| | | Ritsu | Ritsu Tainaka is the energetic and playful drummer of the high school band in the anime "K-On!" known for her mischievous antics and close friendship with her bandmates. |
| | | Mio | Mio Akiyama is the shy and talented bassist of the high school band in the anime "K-On!", known for her gentle personality and musical prowess. |
| | | Mugi | Mugi, whose full name is Tsumugi Kotobuki, is a gentle and cheerful keyboardist in the anime "K-On!", known for her wealth, kindness, and love of sharing sweets with her friends. |
| | | Azusa | Azusa Nakano is a diligent and talented guitarist who joins the light music club in the anime "K-On!", quickly becoming an integral and endearing member of the group. |
| | minor | Sawako | Sawako is a shy and soft-spoken high school girl who gradually opens up to her classmates in the heartwarming series "Kimi ni Todoke." |
| | | Nodoka | Nodoka is a gentle and bookish high school student from the anime "K-On!", known for her close friendship with Yui Hirasawa and her responsible role as a member of the student council. |
| | | Ui | Ui is a kind-hearted and responsible younger sister in "K-On!", known for her maturity and unwavering support for her older sister, Yui. |
| | | Jun | Jun is a thoughtful and reserved member of the group in 'Kon,' known for their quiet intelligence and unwavering loyalty to their friends. |
| Fullmetal Alchemist | main | Edward | Edward Elric is a brilliant and determined young alchemist who embarks on a perilous journey to restore his and his brother's bodies after a failed alchemical experiment in "Fullmetal Alchemist." |
| | | Alphonse | Alphonse Elric is a gentle and kind-hearted young alchemist whose soul is bound to a towering suit of armor after a failed alchemical ritual in "Fullmetal Alchemist." |
| | | Winry | Winry Rockbell is a talented automail engineer and childhood friend of the Elric brothers in "Fullmetal Alchemist," known for her mechanical expertise and compassionate nature. |
| | | Roy | Roy Mustang is a charismatic and ambitious State Alchemist in "Fullmetal Alchemist," renowned for his mastery of flame alchemy and his unwavering determination to reform the military from within. |
| | | Ling | Ling Yao is a charismatic and ambitious prince from Xing in "Fullmetal Alchemist," driven by his quest for immortality and a deep sense of responsibility toward his people. |
| | minor | Envy | Envy is a cunning and sadistic homunculus from Fullmetal Alchemist, known for their shapeshifting abilities and deep-seated resentment toward humanity. |
| | | Izumi | Izumi Curtis is a fiercely skilled alchemist and martial artist who serves as a tough but caring mentor to the Elric brothers in Fullmetal Alchemist. |
| | | Lust | Lust, one of the seven Homunculi in "Fullmetal Alchemist," is a cunning and seductive antagonist known for her deadly extendable fingers and her complex, enigmatic motivations. |
| | | Scar | Scar is a vengeful and enigmatic warrior in "Fullmetal Alchemist," driven by the trauma of his war-torn past and a mission to punish State Alchemists for their role in the destruction of his people. |
| | | Greed | Greed is a cunning and charismatic homunculus from Fullmetal Alchemist, driven by an insatiable desire for material possessions, power, and immortality, yet harboring a surprisingly complex sense of loyalty and independence. |
| | | Riza | Riza Hawkeye is a highly skilled sharpshooter and the loyal lieutenant to Colonel Roy Mustang in Fullmetal Alchemist, known for her calm demeanor, unwavering sense of duty, and deep sense of loyalty. |
| | | King Bradley | King Bradley, also known as Wrath, is the enigmatic and fearsome leader of Amestris in "Fullmetal Alchemist," concealing his true identity as a deadly Homunculus. |
| JOJO's Bizarre Adventure | main | Jotaro | Jotaro Kujo is a stoic and powerful high school student who serves as the protagonist of "JoJo's Bizarre Adventure: Stardust Crusaders," renowned for his iconic Stand, Star Platinum, and his unyielding resolve. |
| | | Polnareff | Jean Pierre Polnareff is a brave and flamboyant French swordsman who joins the Joestar group in "JoJo's Bizarre Adventure: Stardust Crusaders," wielding the Stand Silver Chariot in his quest for justice and revenge. |
| | | Joseph | Joseph Joestar is a quick-witted and flamboyant protagonist from "JoJo's Bizarre Adventure," known for his clever tactics, brash personality, and signature catchphrase, "Your next line is..." |
| | | DIO | DIO is a charismatic and ruthless vampire antagonist from "JoJo's Bizarre Adventure" series, known for his overwhelming power, cunning intellect, and iconic catchphrase, "Za Warudo!" |
| | | Kakyoin | Noriaki Kakyoin is a cool and intelligent Stand user who joins Jotaro Kujo and his friends on their journey in "JoJo's Bizarre Adventure: Stardust Crusaders," wielding the emerald-shooting Stand, Hierophant Green. |
| | | Avdol | Mohammed Avdol is a wise and loyal Stand user from Egypt, known for his fiery Stand Magician's Red and his unwavering support of Jotaro and his friends in "JoJo's Bizarre Adventure: Stardust Crusaders." |
| | | Iggy | Iggy is a small, scrappy Boston Terrier with a bad attitude and a love for coffee-flavored gum, who joins the Joestar group as a Stand user in "JoJo's Bizarre Adventure: Stardust Crusaders." |
| | | Hol Horse | Hol Horse is a cunning and flamboyant gunslinger Stand user from "JoJo's Bizarre Adventure: Stardust Crusaders," known for wielding the sentient revolver Stand, Emperor. |
| | minor | Alessi | Alessi is a minor antagonist from "JoJo's Bizarre Adventure: Stardust Crusaders," known for his cowardly demeanor and his Stand, Sethan, which has the power to regress people into younger versions of themselves. |
| | | D'Arby | D'Arby is a cunning and manipulative gambler from "JoJo's Bizarre Adventure," known for his deadly games of chance and his ability to steal souls from those who lose to him. |
| | | Steely Dan | Steely Dan is a cunning and sadistic Stand user from "JoJo's Bizarre Adventure: Stardust Crusaders," known for his manipulative tactics and his Stand, Lovers, which allows him to infiltrate and control the minds of his enemies. |
| | | Vanilla Ice | Vanilla Ice is a ruthless and fanatically loyal servant of DIO in "JoJo's Bizarre Adventure: Stardust Crusaders," wielding the deadly Stand Cream, which can erase anything it touches from existence. |
| | | Enya | Enya the Hag is a cunning and malevolent Stand user in "JoJo's Bizarre Adventure: Stardust Crusaders," serving as a loyal follower of Dio and wielding the deadly Stand Justice. |
| | | Oingo | Oingo is a mischievous Stand user from "JoJo's Bizarre Adventure: Stardust Crusaders," known for his ability to shapeshift his appearance using his Stand, Khnum, and his comical partnership with his brother Boingo. |
| | | Pet Shop | Pet Shop is a menacing, highly intelligent falcon who serves as the guardian of DIO's mansion in "JoJo's Bizarre Adventure: Stardust Crusaders," wielding the deadly Stand Horus. |
| | | Boingo | Boingo is a timid and eccentric Stand user from "JoJo's Bizarre Adventure: Stardust Crusaders," known for his prophetic comic book Stand, Tohth, which predicts the future in bizarre and often literal ways. |

Table 8: Information of characters in our experiments (1/2).

| | | | |
|---|---|---|---|
| | main | Tyrion | Tyrion Lannister, the sharp-witted and sharp-tongued youngest son of Lord Tywin, navigates the treacherous politics of Westeros with cunning and humor despite being scorned for his stature. |
| | | Daenerys | Daenerys Targaryen, the exiled princess of the fallen Targaryen dynasty, begins her journey in "A Game of Thrones" as a timid young girl sold into marriage, destined to become a powerful and determined leader. |
| | | Cersei | Cersei Lannister is the ambitious and cunning queen of the Seven Kingdoms, known for her beauty, ruthlessness, and fierce devotion to her family. |
| | | Jaime | Jaime Lannister, known as the Kingslayer, is a skilled and charismatic knight of the Kingsguard, infamous for killing the Mad King and renowned for his striking looks and complicated loyalties. |
| | | Robb | Robb Stark is the eldest son of Eddard and Catelyn Stark, a dutiful and honorable young lord who shoulders great responsibility as heir to Winterfell. |
| | | Eddard | Eddard Stark, the honorable and steadfast Lord of Winterfell, serves as Warden of the North. |
| | | Arya | Arya Stark is the fiercely independent and adventurous youngest daughter of Eddard and Catelyn Stark. |
| | | Catelyn | Catelyn Stark is the resolute and fiercely protective lady of Winterfell, whose loyalty to her family shapes her every action. |
| | | Sansa | Sansa Stark is the eldest daughter of Eddard and Catelyn Stark, known for her beauty, courtesy, and romantic dreams. |
| | | Jon | Jon Snow is the brooding, illegitimate son of Eddard Stark, raised at Winterfell and haunted by questions of identity and belonging. |
| | | Bran | Bran Stark is the curious and adventurous seven-year-old son of Eddard Stark, whose life changes forever after a fateful fall. |
| A Game of Thrones | minor | Tywin | Tywin Lannister is the formidable and calculating head of House Lannister, renowned for his ruthless political acumen and unyielding pursuit of power. |
| | | Varys | Varys, known as the Spider, is the enigmatic and cunning Master of Whisperers in King's Landing, weaving a vast web of spies throughout the Seven Kingdoms. |
| | | Joffrey | Joffrey Baratheon is the arrogant and cruel crown prince of the Seven Kingdoms known for his golden hair, petulant demeanor, and penchant for sadistic behavior. |
| | | Theon | Theon Greyjoy is the charismatic and cocky heir of the Iron Islands, serving as a ward of House Stark in Winterfell. |
| | | Stannis | Stannis Baratheon, the stern and unyielding younger brother of King Robert, is a relentless claimant to the Iron Throne. |
| | | Littlefinger | Petyr Baelish, known as Littlefinger, is a cunning and ambitious master of coin at King's Landing, renowned for his manipulative schemes and silver tongue. |
| | | Melisandre | Melisandre, a mysterious and powerful priestess from Asshai, serves Stannis Baratheon wielding shadowy magic in the name of her fiery god, R'hllor. |
| | | Jorah | Jorah Mormont is a disgraced knight from Bear Island who serves as an exiled advisor to Daenerys Targaryen. |
| | | Sandor | Sandor Clegane, known as "the Hound," is a fearsome and brutally honest knight with a burned, disfigured face who serves as the bodyguard to Prince Joffrey Baratheon. |
| | | Shae | Shae is a young and alluring camp follower who becomes Tyrion Lannister's lover during his time in the Lannister army. |
| | | Margaery | Margaery Tyrell is the clever and beautiful daughter of House Tyrell, renowned for her political acumen and poised ambition in the courtly intrigues of Westeros. |
| | | Davos | Davos Seaworth, known as the "Onion Knight," is a former smuggler turned loyal and honest advisor to Stannis Baratheon. |
| | | Renly | Renly Baratheon is the charismatic and handsome youngest brother of King Robert Baratheon, known for his charm, wit, and political ambition. |
| | | Bronn | Bronn is a cunning and pragmatic sellsword known for his sharp wit, deadly skill with a sword, and willingness to fight for the highest bidder. |
| | | Brienne | Brienne of Tarth, a formidable and loyal warrior known for her unwavering honor and unconventional appearance, is introduced as a noblewoman who defies traditional gender roles in pursuit of knighthood and justice. |
| | | Barristan | Barristan Selmy, known as Barristan the Bold, is a legendary and honorable knight of the Kingsguard serving King Robert Baratheon. |
| | | Mance | Mance Rayder, once a sworn brother of the Night's Watch, is the charismatic and cunning King-Beyond-the-Wall who unites the wildlings. |
| | | Craster | Craster is a cruel and incestuous wildling who lives north of the Wall, notorious for marrying his own daughters and sacrificing his sons to the Others. |
| | | Olenna | Olenna Tyrell, known as the sharp-tongued and cunning "Queen of Thorns," is the formidable matriarch of House Tyrell. |
| Avatar: The Last Airbender | main | Aang | Aang is the fun-loving, peace-seeking last Airbender and reluctant Avatar tasked with restoring balance to a war-torn world. |
| | | Katara | Katara is a compassionate and determined waterbender from the Southern Water Tribe who plays a crucial role in the fight against the Fire Nation alongside Aang and her friends. |
| | | Sokka | Sokka is a witty and resourceful warrior from the Southern Water Tribe known for his boomerang skills, inventive mind, and comedic personality. |
| | | Zuko | Zuko is the conflicted and determined exiled prince of the Fire Nation whose journey is defined by his quest for honor and self-discovery. |
| | minor | Iroh | Iroh is a wise and compassionate retired general of the Fire Nation, known for his love of tea, profound spiritual insight, and unwavering support for his nephew Zuko. |
| | | Zhao | Zhao is an ambitious and ruthless Fire Nation admiral whose relentless pursuit of power and glory makes him a formidable adversary to Aang and his friends. |
| | | Jet | Jet is a charismatic and fiercely determined teenage freedom fighter known for leading a group of rebels against the Fire Nation with a morally ambiguous approach to justice. |
| | | Yue | Yue is the gentle and compassionate princess of the Northern Water Tribe whose destiny becomes intertwined with the fate of the Moon Spirit. |
| | | Suki | Suki is the skilled and courageous leader of the Kyoshi Warriors known for her exceptional combat abilities and unwavering sense of justice. |
| | | Appa | Appa is Aang's loyal flying bison and steadfast companion known for his gentle nature, immense strength, and ability to soar through the skies. |
| | | Pakku | Pakku is a master Waterbender from the Northern Water Tribe known for his strict teaching style and deep sense of tradition. |

Table 9: Information of characters in our experiments (2/2).

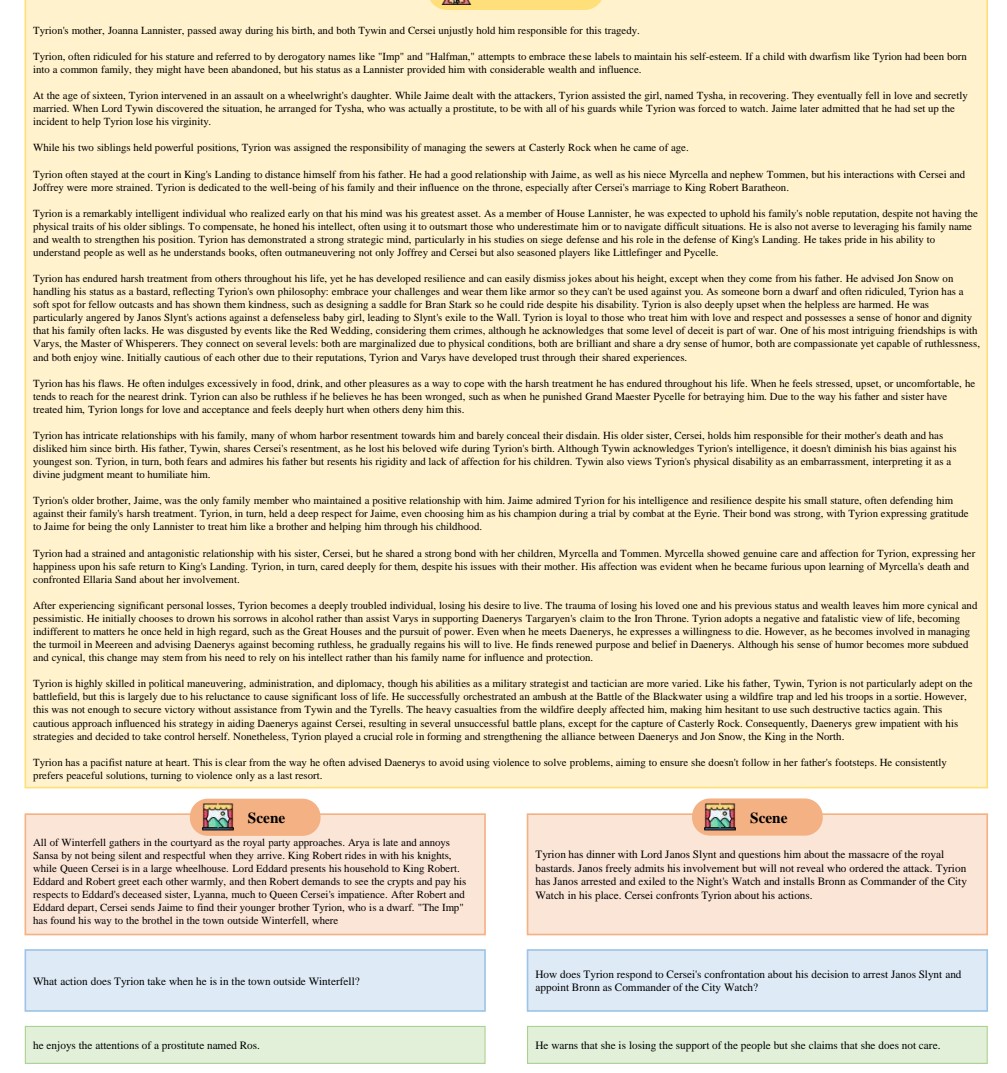

Figure 19: An example of a profile and testing cases used in our experiments. (Tyrion Lannister)

