# OpenReview forum: "Codifying Character Logic in Role-Playing"
_NeurIPS.cc/2025/Conference — NeurIPS 2025 poster_

### Official Review · Reviewer_cxgP · 2025-06-16

**Clarity:** 3
**Significance:** 3
**Originality:** 3
**Rating:** 4
**Confidence:** 4

**Summary:**

This paper proposes a novel method, Codified Profiles, for modeling character logic in LLM-driven role-playing scenarios. The approach converts textual character profiles into executable functions that enable persistence, updatability, and controllable randomness.
A new benchmark, curated from 83 characters and over 5,000 scenes sourced from Fandom, is used for evaluation. Experiments show codified profiles outperform traditional textual approaches in behavioral consistency, adaptability, and diversity, especially when used with smaller LLMs.

**Questions:**

1. In line 248, the reference to Table 6 appears to be incorrect; it should likely be Table 2.
2. Several tables (e.g., Figures 5, 6, and 7) would benefit from clearer labeling of both rows and columns to clarify their meanings. Adding explicit axis labels or captions would improve readability.
3. Similar to Weaknesses 2, how well do codified profiles generalize to unscripted or human-character interactive scenarios beyond the Fandom benchmark?
4. Why is LLM-based NLI used as the main scoring method? Prior works [1-3] have introduced various evaluation strategies—some of which offer finer-grained or more interpretable feedback. What advantages justify choosing NLI over those methods?

[1] Character-LLM: A Trainable Agent for Role-Playing, EMNLP 2023.\
[2] Large Language Models are Superpositions of All Characters: Attaining Arbitrary Role-play via Self-Alignment, ACL 2024.\
[3] MMRole: A Comprehensive Framework for Developing and Evaluating Multimodal Role-Playing Agents, ICLR 2025.

**Ethical Concerns:**

["NO or VERY MINOR ethics concerns only"]

**Final Justification:**

The author has resolved most of my concerns, including those regarding Reliance on Closed LLMs’ Coding Ability, Unscripted Role-playing Performance, Criterion Selection, and Writing Issues. I believe this is an interesting piece of work that deserves to be accepted. However, given that the method has limited evaluation and improvement in common unscripted scenarios, I will maintain my score of borderline accept for this paper.

**Limitations:**

1. Abstraction vs. Fidelity Trade-off: Codification simplifies logic but may lose narrative nuances present in free-form text.

**Paper Formatting Concerns:**

No paper formatting concerns.

**Quality:**

3

**Strengths And Weaknesses:**

**Strengths:**
1. Conceptual Novelty: Introducing executable code as character profiles is a significant and original contribution, effectively combining symbolic logic with LLM capabilities.
2. Controllable Randomness: Explicit modeling of randomness via code enhances reproducibility and behavioral fidelity.
3. Clear Advantages Over Baselines: Demonstrates improvements in persistence, updatability, and behavioral diversity compared to previous methods. The use of a comprehensive benchmark and comparisons across multiple conditions (baseline, evolving, stochastic) strengthen the claims.

**Weaknesses:**
1. Dependence on Proprietary LLMs: GPT-4.1 is used for codification and NLI scoring, raising concerns about reproducibility and transparency. Can the codification pipeline be fully automated using only open-source models, and if so, what trade-offs would arise?
2. Generalization to Open-ended Scenarios: While effective on benchmarked or scripted contexts, the method's ability to generalize to open-ended, unscripted, or human-character interactive settings remains underexplored. As mentioned in Lines 117-121, the rigidity of code may hinder generalization, which is important for role-playing LLMs.

---

> ### Author Rebuttal · Authors · 2025-07-30
>
> We appreciate your positive comments on our work and recognition of our contributions to expand the LLM's ability! In response to your efforts in providing constructive suggestions, we will make the following modifications and clarifications accordingly.
>
> **Reliance on Closed LLM’s Coding Ability (W1)**
>
> We appreciate the reviewer’s concern regarding the reliance on proprietary models such as `gpt-4.1` for codification, especially in terms of reproducibility and transparency. While it is true that closed models (e.g., Claude) currently lead the field in coding tasks, we emphasize that this does not preclude effective codification with open-source LLMs.
> To address this, we conducted new experiments with three recent state-of-the-art open-source LLMs: “moonshotai/Kimi-K2-Instruct” (`kimi-k2`), “Qwen3-235B-A22B-Instruct-2507” (`qwen3-instruct`), and “Qwen3-Coder-480B-A35B-Instruct” (`qwen3-coder`).
>
> |           |                  | Haruhi | K-On! | JOJO  | FMA   | AGOT  | ATLA  | Average   |
> | --------- | ---------------- | ------ | ----- | ----- | ----- | ----- | ----- | --------- |
> | **Main**  | **kimi-k2**          | 74.81  | 67.71 | 62.70 | 68.89 | 67.32 | 64.62 | 67.68     |
> |           | **qwen3-instruct**   | 76.24  | 67.89 | 63.80 | 70.27 | 67.46 | 64.23 | 68.32     |
> |           | **qwen3-coder**      | 74.25  | 69.82 | 63.62 | 70.62 | 67.88 | 69.94 | 69.36     |
> |           | gpt-4.1-mini     | 73.33  | 68.58 | 62.71 | 68.07 | 67.03 | 68.54 | 68.04     |
> |           | gpt-4.1          | 72.14  | 69.23 | 63.92 | 69.60 | 67.51 | 67.90 | 68.38     |
> |           | claude-3.7-sonet | 76.86  | 68.53 | 63.35 | 70.82 | 68.69 | 68.73 | 69.50     |
> |           | claude-4.0-sonet | 75.07  | 68.99 | 63.85 | 74.26 | 67.22 | 70.82 | 70.04 |
> | **Minor** | **kimi-k2**          | N/A    | 67.70 | 65.79 | 65.53 | 71.70 | 64.09 | 66.96     |
> |           | **qwen3-instruct**   |        | 69.48 | 66.08 | 65.64 | 71.71 | 65.70 | 67.72     |
> |           | **qwen3-coder**      |        | 71.63 | 67.51 | 64.44 | 73.64 | 72.06 | 69.86     |
> |           | gpt-4.1-mini     |        | 71.65 | 64.66 | 70.35 | 69.77 | 70.43 | 69.37     |
> |           | gpt-4.1          |        | 68.59 | 70.06 | 68.21 | 70.51 | 71.95 | 69.87     |
> |           | claude-3.7-sonet |        | 75.05 | 67.05 | 67.04 | 73.79 | 69.25 | 70.44     |
> |           | claude-4.0-sonet |        | 71.79 | 67.66 | 70.94 | 74.63 | 73.72 | 71.75 |
>
> As shown in the table above (updated Table 6 in Appendix G), open-source LLMs can rival or even surpass closed models on the codification task, as `qwen3-coder` notably outperforms `gpt-4.1` on main characters and shows only a 0.01 gap on minor characters. This demonstrates that the entire pipeline can be automated with open-source models, significantly improving reproducibility and transparency. We will also release the codification results from closed/open LLMs for reproduction.
>
> While open-source models may have some limitations compared to the very latest proprietary coders (`claude-4.0-sonet`), such as occasional differences in logic abstraction or error rates, our results suggest that these trade-offs are narrowing rapidly. We will update our paper with these findings to further support the viability of open codification models. Thank you for the valuable suggestion!
>
> **Unscripted Role-playing Performance (W2 & Q3)**
>
> We thank the reviewers for highlighting the importance of generalization to open-ended and unscripted scenarios. To address this, we have already included an open-ended character-character interaction setup in Appendix D, where two characters interact in a script-free, unsupervised manner. These results demonstrate that codification can indeed contribute to character consistency and believability even outside benchmarked contexts.
>
> We did not place these results in the main content because reference-based scoring, which is grounded in real, audience-accepted responses, is inherently more reliable than evaluating solely based on LLM judgments of scenario plausibility. We also recognize the value of further exploring human-character interactions. Following the reviewer's suggestion, we conducted additional experiments where a simulated human interviewer (“You are a human interviewer interested in {character}”) driven by `gpt-4.1` interacts unscripted with each character, similar to the methodology in [1]. We ran 83 such interview sessions (1 for each character), and the codified profile achieved 37.35% win, 34.94% tie, and 27.71% loss rates for 8B `llama-3.1` judged by `gpt-4.1`, further validating the generalizability and advantage of codification in broader, open-ended settings. Thank you for raising this important point; we will clarify and update these results in the revised version.
>
> **Criterion Selection (Q4)**
>
> We thank the reviewer for highlighting the diversity of evaluation strategies in recent works and for prompting us to clarify the advantages of our NLI-based approach.
>
> While prior studies such as [1] and [2] utilize multi-dimensional scoring (e.g., emotion, knowledge, value alignment), these typically use the character profile itself as the reference or gold standard for evaluation. This approach is fundamentally different from ours: our benchmark leverages real, audience-accepted plot responses as ground truth, using the profile only as a grounding input rather than as an evaluation reference. In this setting, applying multi-dimensional criteria is not always directly meaningful for each test case, making a unified NLI criterion more appropriate and interpretable for our experiments.
>
> With respect to [3], although it does not input the character profile directly in the evaluation prompt, its methodology still relies on the LLM’s own understanding or "internalization" of the character as the ground truth, which can introduce ambiguity or drift from intended characterization. In contrast, our use of NLI is strictly grounded in concrete plot responses, avoiding circular reasoning.
> NLI further offers practical advantages in granularity and clarity: its three-way decision (entailment, neutral, contradiction) allows for nuanced distinctions  (capturing ambiguous or partially matching responses) without relying on arbitrary numeric scoring or less interpretable axes. When finer granularity is needed (e.g., to specifically evaluate emotional depth), we filter our test set for references mentioning the relevant trait or ability, so NLI can serve as an ability-specific benchmark without altering the core evaluation paradigm. In the following table, we illustrate the filtered (by questioning `gpt-4.1`) performance evaluated by references that include details of “value”, “knowledge”, “emotion”, and “superpower”.
>
> | Dimension | Emotion | Knowledge | Value | Superpower |
> |----------------|-------------|-------------|-------------|-------------|
> |Textual        | 66.19 | 65.76 | 65.82 | 68.66 |
> |Codified      | 68.73 | 69.07 | 69.25 |  72.41|
> |$\Delta$ Score| 2.66 | 3.31 | 3.43 | 3.85 |
> *(Explanation: Metric is averaged over all filtered test cases)*
>
> The result is consistent with our observation that emotion is harder to codify, while superpowers, generally designed to follow audience-understandable logic, benefit more from codification. Thus, NLI can serve as a fine-grained evaluation metric as well, and we will update the results in the paper.
>
> Empirically, we also observe that the LLM NLI discriminator is sensitive to nuanced aspects such as emotion: for example, if a response says "the character willingly helps the antagonist" but the reference says "the character reluctantly helps the antagonist," the NLI correctly detects a contradiction, reflecting awareness of emotional nuance.
>
> Finally, as a well-established and widely used criterion in NLP, NLI ensures that our results are interpretable, reproducible, and comparable with previous literature.
>
> In summary, while multi-dimensional scoring is valuable, our use of NLI is a practical, rigorous, and principled choice for evaluating plot-driven character responses in our benchmark, especially given the design differences from recent alternatives.
>
> **Writing Issues (Q1 & Q2)**
> Thank you for pointing out these writing and formatting issues! We will correct the reference in line 248 to Table 2 and revise the labeling and captions for Figures 5-7 to ensure clearer row and column descriptions (e.g., adding "NLI scores" labels to the y-axis). These updates will be reflected in the revised version.
>
> [1] Character-LLM: A Trainable Agent for Role-Playing, EMNLP 2023.
>
> [2] Large Language Models are Superpositions of All Characters: Attaining Arbitrary Role-play via Self-Alignment, ACL 2024.
>
> [3] MMRole: A Comprehensive Framework for Developing and Evaluating Multimodal Role-Playing Agents, ICLR 2025.

---

> > ### Comment · Reviewer_cxgP · 2025-08-03
> >
> > Thank you for your clarifications. This is an interesting work.

---

### Official Review · Reviewer_YbfS · 2025-06-27

**Clarity:** 3
**Significance:** 3
**Originality:** 4
**Rating:** 5
**Confidence:** 3

**Summary:**

This paper presents codified profiles as a way to use LLMs as in role-playing scenarios with more logic persistence, updatability, and controllable randomness. Codified profiles represents character logic as structured, semi-executable functions for behavioral decision-making. The paper also introduces a new role-playing benchmark and demonstrate the advantage of their approach.

**Questions:**

* Why are eqns (1) and (2) $r = LLM(s | ..)$? Do you mean that the response is sampled from the conditional distribution? $r \sim LLM(\cdot | s, ...)$
* What is the y-axis in Figure 5? Is it still NLI scores? If so, how does different profile representations having similar relative performance in different seasons illustrate "how evolving profiles maintain alignment with narrative shifts" (L280)? Wouldn't that require both (a) a ground-truth line plotted on the same graph showing the same relative performance across seasons, and (b) some assurance that relative NLI differences measure narrative shifts?

**Ethical Concerns:**

["NO or VERY MINOR ethics concerns only"]

**Final Justification:**

After clarifications and added details and reviewing the authors' conversations with other reviewers, I maintain my score.

**Limitations:**

yes

**Quality:**

3

**Strengths And Weaknesses:**

**Good idea, good execution.** The idea of using structured representations to maintain character logic in role-playing settings is interesting, and the results show that it helps produce character responses that are logically consistent with ground-truth responses. The authors run appropriate experiments, evaluation, and analysis. The evaluation dataset is fairly extensive, and seems less prone to data contamination since it is fan fiction. The ablation plots are well-done and show the value of different elements of codified profiles: granularity of segmentation, model size, chain-of-thought length, and structure.

If this paper is published or otherwise publicly shared, I would share it with other researchers interested in structured representations of human logic.

**NLI vs. human preferences.** From the main body of text, it is not immediately clear that NLI against a ground-truth response is the correct measure for free-form LLM response generation. The authors include validation for using NLI as a quality metric and the human preference results in the appendix. I think that especially given the soft-correctness nature of this setting, these should be moved into the main body of text.

---

> ### Author Rebuttal · Authors · 2025-07-30
>
> It's our pleasure to receive your strong support for the acceptance and recommendation to other researchers! To polish our paper to further meet your expectations, we will make the following clarifications and modifications according to your suggestions.
>
> **NLI Validation Placement (W1)**
>
> We agree that human validation is especially important given the potentially noisy nature of LLM evaluation. We will move the human preference results and our validation of NLI as a quality metric from the appendix into the main body of the paper to highlight their importance and improve the paper’s readability.
>
> **Response Sampling Formula (Q1)**
>
> Regarding the notation in equations (1) and (2), our current formulation treats LLM($\cdot$) as a function. We agree with you that representing the response as sampled from a conditional distribution (r ~ LLM($\cdot$| s, ...)) is more precise and will update the notation accordingly.
>
> **Interpretation of Tyrion’s Case Study Result (Q2)**
>
> For Figure 5, the y-axis represents NLI scores (averaged over different seasons). Our claim about evolving profiles maintaining alignment with narrative shifts is based on comparing the performance gap between using static versus evolving codified profiles (the difference between the green solid/dashed line in the right subfigure). This difference becomes most pronounced when there are significant character changes in the narrative (such as Tyrion in AGOT Season 3/4), which we interpret as evidence of the benefit of evolving profiles. We will clarify both the axis labeling and our reasoning in the revised text to make this conclusion easier to follow.

---

> > ### Comment · Reviewer_YbfS · 2025-08-02
> >
> > Thank you for the clarifications. This is an interesting and sound piece of research, and I look forward to seeing the revised manuscript.

---

### Official Review · Reviewer_YKSi · 2025-07-01

**Clarity:** 3
**Significance:** 2
**Originality:** 3
**Rating:** 4
**Confidence:** 4

**Summary:**

This paper introduces Codified Profiles, which aim to represent character logic in role-playing scenarios using structured, executable functions. The primary motivation is to enhance consistency, controllability, and efficiency compared to traditional prompt-based approaches. These profiles convert textual character descriptions into executable functions that include explicit control structures (like if-then-else) and conditional checks handled via helper functions. The approach provides persistence, facilitates systematic updates, and supports controlled randomness in character behaviors. To validate the approach, a new benchmark (Fandom Benchmark) sourced from manga, novels, and television series was developed. Experimental results demonstrate that Codified Profiles improve consistency, adaptability, and diversity of behaviors. Furthermore, it was shown that even small language models like 1B models could effectively use these profiles, making role-playing feasible for lightweight deployments.

**Questions:**

Please see weaknesses above.

**Ethical Concerns:**

["NO or VERY MINOR ethics concerns only"]

**Limitations:**

yes

**Quality:**

3

**Strengths And Weaknesses:**

Strengths
- The concept of transforming textual character logic into structured, executable code is innovative and effectively addresses current challenges in consistency and interpretability in LLM-driven role-play.
- Experiments clearly validate key claims about persistence, controllable randomness, and ease of updating character profiles. The new Fandom Benchmark can be useful for future research in the field of LLM for role-playing.



Weaknesses
- The method relies on LLMs (e.g., GPT-4.1) to convert text to code, which may introduce errors in complex logic.
- NLI scoring (entailment/neutral/contradiction) lacks granularity for nuanced role-playing quality (e.g., emotional depth).

---

> ### Author Rebuttal · Authors · 2025-07-30
>
> We are grateful to see your positive and valuable feedback on our work! On your concern about the potential error introduction and the criterion selection, we make the following clarifications and modifications accordingly. We believe our work is further polished with your constructive suggestions!
>
> **Logic Interpretation from Text v.s. Code (W1)**
>
> We completely understand the concern that codifying text into code may introduce errors, especially when handling complex logic. However, we also would like to emphasize that such errors are more likely to occur when relying solely on LLMs to interpret textual profiles directly during role-playing. Correct profile interpretation is equally essential when role-playing with textual profiles, as the LLM must extract and apply relevant logic in real time, which can be problematic and inconsistent. One core motivation for codification is to reduce cases where “the LLM can interpret the logics in the profile but fails to execute them consistently during role-play.”
>
> Additionally, codification makes the interpreted logic explicit and thus human-supervisable. Errors in codified profiles can be identified and debugged systematically, whereas errors arising from on-the-fly reasoning with textual profiles are often impersistent/opaque and hard to diagnose. In this sense, while codification may introduce errors, those errors are debuggable and correctable, unlike the implicit reasoning failures of direct textual interpretation. As the potential error in codification is also likely to occur in on-the-fly reasoning, it’s better to attribute the introduced error to the LLM’s interpretation rather than the codification mechanism.
>
> As codification is an extra preprocessing step in role-playing, it's natural for readers to have error introduction concerns about integrating this codification system. We will incorporate the discussions above into our paper to further clarify the position of codification. Many thanks for pointing out this potential concern!
>
> **Criterion Selection (W2)**
>
> We thank the reviewer for highlighting the importance of multi-dimensional evaluation in role-playing. We would like to explain the selection of NLI criterion based on **reference difference**,  **granularity**, **adaptability**, and **connection to NLP** in our work.
>
> **(Reference Difference)** While previous works (e.g. literature [1-3] mentioned by reviewer cxgP) often rate responses across multiple axes (e.g., emotion, knowledge, value alignment), these works typically use the character profile (or LLM's internal understanding of the profile) itself as the reference. In contrast, our benchmark utilizes real, audience-accepted plot responses as ground truth, with the profile serving only as a grounding input. In this setting, multi-dimensional criteria are not always directly applicable to the reference in each test case because not all dimensions will exist in the reference, making a unified NLI criterion more appropriate for our experiments.
>
> **(Granularity)** NLI also provides useful granularity by introducing a neutral state, capturing partial or ambiguous matches more reliably than binary yes/no judgments. NLI also offers a clear, well-defined criterion than numeric scoring (e.g., rating 1-10) as well.
>
> **(Adaptability)** When finer evaluation granularity is needed (e.g., evaluating emotional depth), we can filter our test set for reference responses that mention a target ability (e.g., emotion states), allowing NLI to serve as a fine-grained, ability-specific benchmark without altering our core evaluation method. In our validation of the LLM NLI discriminator, we observe that it takes every detail in the reference (including emotion) into account. For example, when the response says “the character willingly helps the antagonist” while the reference says “the character reluctantly helps the antagonist”, it will be discriminated as “contradicted”, showing the awareness of emotional depth. In the following table, we illustrate the filtered (by questioning gpt-4.1) performance evaluated by references that include details of “value”, “knowledge”, “emotion”, and “superpower”.
>
> | Dimension | Emotion | Knowledge | Value | Superpower |
> |----------------|-------------|-------------|-------------|-------------|
> |Textual        | 66.19 | 65.76 | 65.82 | 68.66 |
> |Codified      | 68.73 | 69.07 | 69.25 |  72.41|
> |$\Delta$ Score| 2.66 | 3.31 | 3.43 | 3.85 |
> *(Explanation: Metric is averaged over all filtered test cases)*
>
> The result is consistent with our observation that emotion is harder to codify, while superpowers, generally designed to follow audience-understandable logic, benefit more from codification. Thus, NLI can serve as a fine-grained evaluation metric as well, and we will update the results in the paper.
>
>
> **(Connection to NLP)** Finally, as a formally established criterion in NLP, NLI ensures our results are interpretable and comparable with prior literature.
>
> In summary, while multi-dimensional scoring has value, our use of NLI provides the most practical and rigorous approach for assessing plot-driven character responses in our benchmark.
>
> [1] Character-LLM: A Trainable Agent for Role-Playing, EMNLP 2023.
>
> [2] Large Language Models are Superpositions of All Characters: Attaining Arbitrary Role-play via Self-Alignment, ACL 2024.
>
> [3] MMRole: A Comprehensive Framework for Developing and Evaluating Multimodal Role-Playing Agents, ICLR 2025.

---

> > ### Comment · Reviewer_YKSi · 2025-08-05
> >
> > Thank you for your comments. It is an interesting work, and these results can be included in the revised version.

---

### Official Review · Reviewer_o66c · 2025-07-02

**Clarity:** 3
**Significance:** 2
**Originality:** 3
**Rating:** 5
**Confidence:** 4

**Summary:**

This paper introduces a novel framework called "Codified Profiles" for LLM-based character role-playing. Instead of appending plain text character descriptions to prompts, this method compiles them into structured, executable Python functions. These functions use explicit control logic (e.g., if-then-else) and a flexible helper function (check_condition) that queries an LLM to interpret scene-specific conditions. The authors argue that this approach offers three main advantages over traditional methods: Persistence (consistent application of character logic), Updatability (easier to debug and evolve character logic over a storyline), and Controllable Randomness (precise simulation of stochastic behaviors). To validate these claims, the paper introduces a new comprehensive benchmark, "Fandom Benchmark," curated from narrative websites. Experiments demonstrate that Codified Profiles improve role-playing consistency, are more adaptable, and allow for greater behavioral diversity, showing particular strength when implemented with smaller language models.

**Questions:**

1.How robust is the codification process to the style of the source profile? For example, a Fandom wiki article is often structured and factual. Would the codification LLM struggle more with a profile written in a more literary or implicit style, potentially leading to less accurate or incomplete logic?

2.The check_condition function is a cornerstone of the framework's flexibility. It can return True, False, or None (Unknown). How does the system handle an "Unknown" response? Could this lead to situations where no logical branch is triggered, resulting in a generic or null response from the character?

3.The evolving mechanism is supervised by ground-truth actions from the storyline. How do you envision this mechanism adapting to an interactive, open-ended role-playing scenario where no such ground truth exists? Could the framework be integrated with self-reflection mechanisms to enable unsupervised character evolution based on interaction history?

4.The examples in the paper show clear but relatively simple logical structures. In your experiments with complex characters from works like "A Game of Thrones", did the codification LLM generate more complex code involving, for instance, state variables that persist across multiple scenes or nested loops? If so, how does the framework manage this statefulness?

**Ethical Concerns:**

["NO or VERY MINOR ethics concerns only"]

**Final Justification:**

The authors' response addressed most of my concerns, but I still believe that executable codes cannot represent most scenarios. Therefore, the method is only applicable to cases with verifiable facts.

**Limitations:**

Yes

**Quality:**

3

**Strengths And Weaknesses:**

Strengths:

1.The paper addresses a well-defined and important problem in LLM-based role-playing: the lack of persistence, controllability, and transparency in prompt-based methods. The proposed solution of "codifying" character logic into executable functions is highly original and offers a new paradigm for character representation that synergizes symbolic logic with neural models. This is a significant departure from mainstream RAG or fine-tuning approaches.

2.The experimental validation is a major strength of this paper. The authors systematically design experiments to verify each of their three core claims:
- Persistence: The superiority of codified profiles over textual profiles and RAG baselines in Table 2 and Figure 4 clearly supports this claim.
- Updatability: The "Evolving Profile" experiment (Section 5.3, Figure 5) provides strong evidence that codified logic is more effectively and efficiently updated along a storyline compared to unstructured text.
- Controllable Randomness: The experiments in Figure 6 convincingly show that codified profiles can simulate specified probabilities accurately, a known weakness of LLM sampling, and that this controlled stochasticity leads to better coverage of plausible actions (Best@K results).

3.By curating 83 characters and over 5,000 scenes from rich, real-world narrative sources, the authors provide a high-quality, behavior-centric benchmark that will be valuable for future research in this area.

4.The paper is exceptionally well-written and easy to follow. The methodology is explained clearly with illustrative examples. The appendices are thorough, providing all necessary prompts, benchmark details, and extended case studies, which strongly supports the reproducibility of the work.

----

Weaknesses:

1.The framework's effectiveness hinges on the ability of a powerful LLM (GPT-4.1 in this case) to accurately translate nuanced text into correct logical code. The performance could degrade substantially if a less capable, open-source model were used for this codification step. While Appendix G explores this, the core experiments rely on a state-of-the-art closed model, which could be a practical barrier.

2.The proposed evolving mechanism is driven by NLI-based comparisons to a ground-truth reference action from the original story. This is a sound approach for evaluation but its applicability to open-ended, interactive role-playing (where no ground-truth exists) is unclear. The paper does not fully explore how character evolution would be triggered or guided in such scenarios.

3.The process of abstraction from rich, descriptive text to structured code may lead to a loss of nuance. A character's personality might contain subtle elements that are not easily captured by discrete if-then logic. The authors acknowledge this in Appendix A and show that combining codified and textual profiles yields slight gains (Table 2), which implicitly supports this concern. This trade-off could be discussed more prominently in the main paper.

---

> ### Author Rebuttal · Authors · 2025-07-30
>
> We are sincerely thankful for your strong support for our work with both strong recommendations and insightful comments/questions. In response to your valuable efforts to further improve our work, we have the following clarifications and modifications. We believe our work is further solidified with your suggestions!
>
> **Reliance on Closed LLM’s Coding Ability (W1)**
>
> We completely agree that the benefit of codified profiles comes from the coding ability of modern LLMs and the growing ability is actually one motivation for our work. While closed coders (e.g., Claude) dominate the state-of-the-art in coding, this does not mean no chance for open-source coders. **We add experiments with three recent state-of-the-art open source LLMs:** “moonshotai/Kimi-K2-Instruct” (`kimi-k2`), “Qwen3-235B-A22B-Instruct-2507”(`qwen3-instruct`), “Qwen3-Coder-480B-A35B-Instruct” (`qwen3-coder`) to Table 6 in Appendix G (codification model comparison).
>
> |           |                  | Haruhi | K-On! | JOJO  | FMA   | AGOT  | ATLA  | Average   |
> | --------- | -------- | ------ | ----- | ----- | ----- | ----- | ----- | --------- |
> | **Main**  | **kimi-k2**          | 74.81  | 67.71 | 62.70 | 68.89 | 67.32 | 64.62 | 67.68     |
> |           | **qwen3-instruct**   | 76.24  | 67.89 | 63.80 | 70.27 | 67.46 | 64.23 | 68.32     |
> |           | **qwen3-coder**      | 74.25  | 69.82 | 63.62 | 70.62 | 67.88 | 69.94 | 69.36     |
> |           | gpt-4.1-mini     | 73.33  | 68.58 | 62.71 | 68.07 | 67.03 | 68.54 | 68.04     |
> |           | gpt-4.1          | 72.14  | 69.23 | 63.92 | 69.60 | 67.51 | 67.90 | 68.38     |
> |           | claude-3.7-sonet | 76.86  | 68.53 | 63.35 | 70.82 | 68.69 | 68.73 | 69.50     |
> |           | claude-4.0-sonet | 75.07  | 68.99 | 63.85 | 74.26 | 67.22 | 70.82 | 70.04 |
> | **Minor** | **kimi-k2**          | N/A    | 67.70 | 65.79 | 65.53 | 71.70 | 64.09 | 66.96     |
> |           | **qwen3-instruct**   |        | 69.48 | 66.08 | 65.64 | 71.71 | 65.70 | 67.72     |
> |           | **qwen3-coder**      |        | 71.63 | 67.51 | 64.44 | 73.64 | 72.06 | 69.86     |
> |           | gpt-4.1-mini     |        | 71.65 | 64.66 | 70.35 | 69.77 | 70.43 | 69.37     |
> |           | gpt-4.1          |        | 68.59 | 70.06 | 68.21 | 70.51 | 71.95 | 69.87     |
> |           | claude-3.7-sonet |        | 75.05 | 67.05 | 67.04 | 73.79 | 69.25 | 70.44     |
> |           | claude-4.0-sonet |        | 71.79 | 67.66 | 70.94 | 74.63 | 73.72 | 71.75 |
>
> The results in the Table above verify that open source LLMs can rival closed ones on the codification task, especially given that `qwen3-coder` outperforms `gpt-4.1` on main characters and has only a 0.01 gap on minor characters. The results indicate the implementation barrier for codification isn’t that high and does not need to rely on closed LLMs, which further improves the reproducibility of our and future codification works. We will update the results in the main content and appendix to show that open source coders can also support codification implementation. Thanks for the suggestion!
>
> **Evolving Codified Profiles in Open-ended Scenarios (W2 & Q3)**
>
> How to evolve codified profiles in open-ended scenarios is an insightful and interesting topic for discussion! In the current version of our work, the evolving mechanism is designed to capture character dynamics in the scripted storyline (e.g., it can provide different Tyrion’s profiles in Season 1 v.s. Season 5 for users to interact with), but characters can certainly evolve in an open-ended environment with interactions. For self-reflection, we agree it’s an important mechanism for accurate self-update when update signals are received. But here we would like to emphasize what can be used as trustworthy update signals when evolving character logics without ground-truth plots. Here we discuss three potential feedback sources: **Environment Feedback**, **User’s Preference**, and **Pre-defined Evolving Mechanism**.
>
> - **Environment Feedback:** Characters can adapt using environmental signals such as action outcomes, changes in relationships, and feedback from other characters. These serve as cues for adjusting motivations, beliefs, and traits.
>
> **Prerequisite:** This requires building a world-level feedback system, which is discussed in our limitation section for future work.
>
> - **User’s Preference (Including authorial intent):** Explicit or implicit user feedback (e.g., corrections, engagement patterns) and narrative intent can guide updates, ensuring evolution aligns with player expectations or desired scenarios.
>
> **Prerequisite:** This requires a man-in-the-loop system to let users tell the role-playing system what kind of character behavior is preferred to update their logic.
>
> - **Pre-defined Evolving Mechanism:** Characters can be initialized with plausible development paths or constraints. Self-reflection modules use interaction history to trigger updates along these paths, preserving coherence without ground-truth supervision.
>
> **Prerequisite:** This requires a self-evolving logic to be written or codified inside the initial profile of the character.
>
> Self-evolving without ground truth is feasible, but it cannot happen without any feedback signals. Reliable update signals from the environment, users, or pre-defined trajectories are essential for meaningful character evolution. We sincerely thank the reviewer for raising this valuable discussion point, which will help us better position our framework for open-ended interactive scenarios. We will update such a future work discussion after the limitation section to further expand the potential future work discussions.
>
> **Nuanced or Implicit Profiles (W3 & Q1)**
>
> We thank the reviewer for raising this insightful discussion on codifying implicit/nuanced profiles. For this question, we think it’s important to discuss the following two cases of implicit expressions.
> - **Case 1 (Implicit expression with hidden logics):** If the implicit or nuanced profiles indeed contain hidden logics that are simply conveyed in a more literary style, these subtleties pose challenges not only to codification but also to directly interpreting textual profiles during role-playing. In such cases, codification can actually help by surfacing and converting these latent logics into more reliable grounding functions, thereby improving consistency.
> - **Case 2 (Implicit expression without hidden logics):** If, instead, the implicit or nuanced profiles lack such hidden logics and instead emphasize tone or descriptive richness, we agree they may be better left uncodified, as discussed in the limitations section. We fully concur that developing a discriminator to assess whether a profile contains sufficient logic depth for codification is an important direction for future work. This would help avoid breaking nuanced information when textual profiles serve primarily stylistic purposes.
>
> We will highlight this trade-off more prominently in the main paper and we appreciate the reviewer for prompting this valuable clarification.
>
> **Handling "Unknown" Branches (Q2)**
>
> Allowing `check_condition` to return `None` (Unknown) prevents the framework from incorrectly defaulting to `False` when it cannot confidently determine the condition. As in the Roy (FMA) example below, if the scene's formality is unclear (`None`), no statement is generated. This avoids unwarranted or generic actions and keeps character behavior consistent with available information.
>
> ```python
>     # 1. Is the setting formal or serious?
>     formal = check_scene(scene, "Is the setting formal or serious?")
>     if formal is True:
>         scene_statements.append("Roy styles his hair neatly, slicking it back for a polished appearance.")
>     elif formal is False:
>         scene_statements.append("Roy's hair is casually tousled, falling over his eyes.")
>     # If unknown, do not add a statement.
> ```
>
> **Complexity and Statefulness (Q4)**
>
> Currently, our codified logic is largely "Markov", which processes each scene independently without persistent state or memory, so we have not observed LLMs producing that complex code (deeply nested condition checking exists, referring to Table 5). However, codification can represent more complex logic, such as loops or persistent state, as illustrated by the “Endless Eight” example below (from Haruhi Suzumiya, where characters repeat the same period until a key action breaks the loop) (This is not simulating the profile but an event so it's not included in our experiments and is codified in a different format). This demonstrates the framework’s potential for handling complex logic (e.g., time loops) if needed.
>
> ```python
> scene_statements = []
> # Does the Endless Eight time loop trigger?
> if check_scene(scene, "Are Haruhi and the SOS Brigade on summer vacation and making plans?"):
>     scene_statements.append("Endless Eight time loop is triggered! The characters are repeating the last two weeks of summer.")
>     n_loop = 1
>     while check_scene(scene, "Is the time loop still repeating?"):
>         scene_statements.append(f"Loop #{n_loop}: The Brigade does summer activities, but feels a sense of déjà vu.")
>         user_action = input("What will you do differently this time? (e.g., 'Suggest doing homework', ...): ")
>         scene = update_scene(scene, scene_statements, user_action ) # Psuedo Implementation
>         if check_scene(scene, "Did someone suggest finishing their summer homework?"):
>             scene_statements.append("Kyon suggests doing his homework. Haruhi approves. The Endless Eight loop is broken!")
>             break
>         elif check_scene(scene, "Is someone aware of the loop but fails to act?"):
>             scene_statements.append("The Brigade feels uneasy, but no decisive action is taken. The loop continues...")
>             n_loop += 1
>         else:
>             scene_statements.append("Nothing significant changes. The loop continues...")
>             n_loop += 1
> ```
> *(Generated by GPT-4.1 based on the description of Endless Eight)*

---

> > ### Comment · Reviewer_o66c · 2025-08-02
> >
> > Thank you for your reply. This is an interesting work that makes a valuable contribution to the role-playing field.

---

### Decision · Program_Chairs · 2025-09-17

**Decision:**

Accept (poster)

**Comment:**

This paper proposes Codified Profiles, a new framework for representing character logic in role-playing by converting textual descriptions into structured, executable functions. Unlike prompt-based methods, codified profiles use explicit control structures and flexible condition checks to produce consistent, updatable, and stochastic behaviors. The authors introduce a benchmark of 83 characters and over 5,000 scenes from Fandom and evaluate persistence, adaptability, and diversity through NLI-based scoring. Results show that codified profiles significantly improve consistency and controllability, and even small models (1B parameters) can perform high-quality role-playing when leveraging this representation.

The paper’s strengths lie in its conceptual novelty, clear exposition, strong experimental validation, and contribution of a valuable benchmark. Reviewers praised the originality of combining symbolic logic with LLM reasoning and confirmed that experiments convincingly support the three core claims. Weaknesses primarily concern reliance on closed LLMs for codification, potential loss of narrative nuance, and questions about generalization to open-ended role-playing. The rebuttal directly addressed these issues: additional experiments showed open-source coders rival closed models, clarifications outlined how evolution could occur in interactive scenarios, and the role of NLI as a principled evaluation metric was justified with supplementary results and planned revisions. Post-rebuttal comments from all reviewers were positive, with concerns adequately resolved. Therefore, I recommend acceptance.